# Black carbon and organic carbon dataset over the Third Pole

Shichang Kang[1,11]*, Yulan Zhang[1]*, Pengfei Chen[1], Junming Guo[1], Qianggong Zhang[2], Zhiyuan Cong[2], Susan Kaspari[3], Lekhendra Tripathee[1], Tanguang Gao[4], Hewen Niu[1], Xinyue Zhong[5], Xintong Chen[1], Zhaofu Hu[1], Xiaofei Li[6], Yang Li[7], Bigyan Neupane[8], Fangping Yan[1], Dipesh Rupakheti[9], Chaman Gul[10], Wei Zhang[1], Guangming Wu[2], Ling Yang[1], Zhaoqing Wang[4], Chaoliu Li[1]

[1] State Key Laboratory of Cryospheric Science, Northwest Institute of Eco-Environment and Resources, Chinese Academy of Sciences, Lanzhou 730000, China

[2] State Key Laboratory of Tibetan Plateau Earth System, Resources and Environment, Institute of Tibetan Plateau Research, Chinese Academy of Sciences, Beijing 100101, China

[3] Department of Geological Sciences, Central Washington University, Ellensburg, Washington, USA

[4] Key Laboratory of Western China's Environmental Systems (Ministry of Education), College of Earth and Environmental Sciences, Lanzhou University, Lanzhou 730000, China

[5] Key Laboratory of Remote Sensing of Gansu Province, Northwest Institute of Eco-Environment and Resources, Chinese Academy of Sciences, Lanzhou, 730000, China

[6] School of Environmental Science and Engineering, Shannxi University of Science and Technology, Xi'an 710021, China

[7] Institute of International Rivers and Eco-security, Yunnan University, Kunming, Yunnan, 650091, China

[8] School of Geography, South China Normal University, Guangzhou 510631, China

[9] Jiangsu Key Laboratory of Atmospheric Environment Monitoring and Pollution Control, Collaborative Innovation Center of Atmospheric Environment and Equipment Technology, School of Environmental Science and Engineering, Nanjing University of Information Science & Technology, Nanjing 210044, China

[10] Reading Academy, Nanjing University of Information Science and Technology, Nanjing, 210044, China

[11] University of Chinese Academy of Sciences, Beijing 100049, China

*Correspondence to: Prof. Shichang Kang (shichang.kang@lzb.ac.cn) and Dr. Yulan Zhang (yulan.zhang@lzb.ac.cn)

**Abstract.** The Tibetan Plateau and its surroundings, also known as the Third Pole, play an important role in the global and regional climate and hydrological cycle. Carbonaceous aerosols (CAs), including black carbon (BC) and organic carbon (OC), can directly or indirectly absorb and scatter solar radiation, and change the energy balance on Earth. CAs, along with the other atmospheric pollutants (e.g., mercury), can be frequently transported over long distances into the inland Tibetan Plateau. During the last decade, a coordinated monitoring network and research program on Atmospheric Pollution and Cryospheric Change (APCC) has been gradually setup and continuously operated within the Third Pole regions to investigate the linkage between atmospheric pollutants and cryospheric change. This paper presents a systematic dataset of BC, OC, water soluble organic carbon (WSOC), and water insoluble organic carbon (WIOC) from aerosols (20 stations), glaciers (17 glaciers, including samples from surface snow/ice, snowpit, and two ice cores), snow cover (2 stations continuous observed, and 138 locations surveyed once), precipitation (6 stations), and lake sediment cores (7 lakes) collected across the Third Pole, based on APCC program. These data were created based on online (in-situ) and laboratory measurements. High-resolution (daily scale) atmospheric equivalent BC concentrations were obtained by using an Aethalometer (AE-33) in the Mt. Everest (Qomolangma) region, which can provide a new insight into the mechanism of BC transportation over the Himalayas. Spatial distributions of BC, OC, WSOC and WIOC from aerosols, glaciers, snow cover, and precipitation indicated different features

among the different regions of the Third Pole, which were mostly influenced by emission sources, transport pathways, and

deposition processes. Historical records of BC from ice cores and lake sediment cores revealed the strength of human activities impacts since the Industrial Revolution. BC isotopes from glaciers and aerosols identified the relative contributions of biomass and fossil fuel combustion to BC deposition on the Third Pole. Mass absorption cross section of BC and WSOC from aerosol, glaciers, snow cover, and precipitation samples were also provided. This updated dataset is released to the scientific communities focusing on atmospheric science, cryospheric science, hydrology, climatology and environmental science. The

related datasets are presented in the form of excel files. These files are available to download from the National Cryosphere Desert Data Center (http://www.ncdc.ac.cn), Northwest Institute of Eco-Environment and Resources, Chinese Academy of Sciences (Kang S et al., 2021. Black carbon and organic carbon dataset over the Third Pole. National Cryosphere Desert Data Center, 2021. https://doi.org/10.12072/ncdc.NIEER.db0114.2021).

**Keywords.** Black carbon, Organic carbon, Aerosol, Glacier, Ice core, Lake sediment cores, Tibetan Plateau


**Abbreviations.**

TP  Tibetan Plateau

BC  black carbon (referred as a general concept; also equivalent to elemental carbon (EC) for the samples of aerosols, lake sediment cores, and glacier surface snow and snowpits; detected by the thermal-optical method using modified protocol)

EC  elemental carbon (aerosol samples measured by the thermal-optical method using the standard IMPROVE_A protocol)

eBC  equivalent black carbon, measured by a real-time optical instrument (Aethalometer model AE-33)

rBC  refractory black carbon (detected by the Single Particle Soot Photometer, SP2)

OC  organic carbon (referred to as total organic carbon from the filtered aerosol samples detected by the thermal-optical method)

WSOC  water soluble organic carbon (referred to dissolved organic carbon in the samples of glacier, snow cover, and precipitation; also

referred to as the water soluble portion of OC from filtered aerosol samples; detected by a total organic carbon analyser)

WIOC  water insoluble organic carbon (referred to as insoluble organic carbon from the filtered glacier, snow cover, and precipitation samples, detected by the thermal-optical method)

MAC  mass absorption cross section

# 1 Introduction

With high elevations (average > 4000 m a.s.l.), the Tibetan Plateau (TP) and its surroundings, known as the Third Pole, plays an important role in the Earth's climate through its complex topography (Yao et al., 2019). Due to the wide distributions of mountain glaciers, snow cover, permafrost and seasonally frozen ground, the TP and its surroundings are also known as the Asian Water Tower (Immerzeel et al., 2010; Yao et al., 2012), which are the source regions of several large Asian rivers (e.g., Yellow, Yangtze, Brahmaputra, Ganges, and Indus rivers). The TP is particularly sensitive to climate change and regional

anthropogenic forcing, and currently has been experiencing significant warming (Chen et al., 2015; Gao et al., 2019; Huss and Hock, 2019; IPCC, 2021; Kang et al., 2010; Ramanathan et al., 2007a; Ramanathan and Carmichael, 2008; Xu et al., 2009; You et al., 2021). Recent rapid cryospheric changes (e.g., glacier melting, permafrost thawing, snow cover declining) on the TP profoundly affect the regional water cycle and ecosystems (Brun et al., 2020; Chen et al., 2019a; Kang et al., 2015; Immerzeel et al., 2019; Nie et al., 2019; Sun et al., 2021; Yao et al., 2012, 2019; Zhang et al., 2020a).

Due to the harsh environment and poor accessibility, observed data on the TP are still scares, which limits the effective investigation, verification, and calibration of reanalysis data and modelling simulations (Qian et al., 2015; You et al., 2020). In particular, studies on atmospheric carbonaceous components (e.g., black carbon, BC; organic carbon, OC) have gained wide attentions, due to the fact that they can directly or indirectly absorb or scatter solar radiation, alternately interact with the nucleation of clouds, and influence the precipitation efficiency (Bond et al., 2013; IPCC, 2013; Ji et al., 2016; Ramanathan et

al., 2005; Ramanathan and Carmichael, 2008; Ramachandran et al., 2020; Yang et al., 2020). These carbonaceous aerosols (CAs) have substantially influenced the climate and environmental changes on Earth (Kang et al., 2020; Li et al., 2017a; Xu et al., 2012; Chen et al., 2019b; Zhang et al., 2017a; Zhang et al., 2012). The TP has recently been polluted by anthropogenic emissions mainly from long-range atmospheric transport, especially during the pre-monsoon when the aerosols accumulated and combined with westerlies and local mountain valley breeze (Cong et al., 2015; Kang et al., 2019; Painicker et al., 2021;

Yang et al., 2018). Although the atmospheric environment over the TP is minimally disturbed by local anthropogenic activities because of the sparse population and limited industries (Cong et al., 2013; Kang et al., 2019), local emissions from booming tourism traffic or domestic fossil fuel/biomass burning also potentially contributed to the climatic and environmental changes (Li et al., 2018a). The investigations of CAs from various environmental media in the TP will improve our understanding of recent environmental changes and their impact on rapid cryospheric melting under climate warming.

The glaciers provide natural archives of climate and environmental information (Jouzel, 2013; Thompson, 2000; Yao et al., 2006). Historical BC records from the Tibetan glacier ice cores revealed the impact of anthropogenic emissions, with BC concentrations increased by 2−3 times since the 1950s (Kaspari et al., 2011; Wang et al., 2015, 2020). Light absorbing BC and OC (including water soluble organic carbon (WSOC) and water insoluble organic carbon (WIOC)) deposited on the glacier and snow cover induced surface darkening and enhanced melting (Flanner et al., 2007; Kang et al., 2020; Lau and Kim, 2018;

Santra et al., 2019; Xu et al., 2009; Zhang et al., 2020b, 2021a). Estimates indicated that BC in snow resulted in accelerated glacier melt by approximately 15−20% in the southeast/central TP and Central Asia (Li et al., 2020; Zhang et al., 2017, 2020b),

and reduced snow cover duration by 3−4 days across the TP and 3−6 days in Northern Xinjiang (Zhang et al., 2018a; Zhong et al., 2019). Meanwhile, WSOC in snow may also contribute to glacier melting (Hu et al., 2018; Gao et al., 2020), due to their light absorption property equal to approximately 10% of that by BC in the northern TP and Northern Xinjiang (Yan et al., 2016; Zhang et al., 2019). The isotopic signatures of BC in glacier and modelling results constrained BC sources in the TP, which indicated that BC was predominately derived from fossil fuel combustion (>60%) in the northern TP. Meanwhile in the southern TP, comparable contributions from fossil fuel combustion (46%) and biomass burning (54%) to BC were observed (Li et al., 2016a; Yang et al., 2018; Zhang et al., 2017a). WSOC in precipitation can provide carbon to the ecosystems (Li et al., 2018b). It was reported that the average WSOC concentrations of precipitation at Nam Co region (inland TP) were lower than that in the urban areas but higher than that from snowpit samples (Li et al., 2018b).

Over the TP, large numbers of lakes are distributed (including proglacial lakes) (Brun et al., 2021; Zhang et al., 2020a). As BC is chemically inert in the lake sediment cores, it can serve as an archive and reliable indicator to investigate the source and transport of BC in the past (Han et al., 2012). The BC investigation from lake sediment cores over the plateau revealed an increasing trend of BC concurrent with increased anthropogenic emissions since the 1950s (Cong et al., 2013; Han et al., 2015; Neupane et al., 2019; Zhu et al., 2020), and suggested that BC deposition in recent decades have increased about 2−3 fold compared to the background level.

During the past decade, our research team has gradually setup a coordinated monitoring network and research program to link Atmospheric Pollution and Cryospheric Change (APCC) covering the TP and its surrounded region (Fig. 1) (Kang et al., 2019). Based on the APCC program, our overarching goal is to perform more integrated and in-depth investigations of the origins and distributions of atmospheric pollutants and their impacts on cryospheric change. The updated specific goals include:

(I) Characterize the features of atmospheric pollutants (including new emergent pollutants) and depict their spatial and seasonal variations in different environmental media over the Third Pole region;

(II) Investigate the source appointment of different atmospheric pollutants based on chemical tracers and modelling, and reveal the transport pathways and mechanisms by which atmospheric pollution is trans-boundary transported to the Third Pole region;

(III) Quantify the contribution of atmospheric pollutants deposited as light-absorbing impurities to the glacier and snow melting, determine the fates of environmentally relevant pollutants within glaciers and snowpack, and further estimate the feedback of cryospehric melting to the carbon (and nitrogen) cycle and hydrological changes.

A series achievement has been published on the research progress that depends on or is related to the APCC program during the past several years (Kang et al., 2019). Recently, new progresses were also achieved, including extended the study area to central Asia (Chen et al., 2021; Zhang et al., 2021a), investigated a new emergent pollutant-microplastics in snow (Zhang et al., 2021b), and discussed the potential impact of glacier melting and permafrost thaw on carbon cycle (Gao et al., 2021a; Zhang et al., 2021c). Therefore, in this article, we will introduce and provide access to the systematic dataset of BC and OC from the atmosphere, glaciers (including ice cores), snow cover, precipitation, and lake sediment cores over the TP and its surroundings based on the APCC program. The site description, online observations, and sampling are summarized in

Section 2 and 3, respectively. A detailed data description of laboratory analysis and BC, OC, WSOC, and WIOC concentrations from aerosols, glaciers, precipitation, snow cover, and refractory BC (rBC) historical records from ice cores and BC from lake sediment cores are given in Section 4, highlighting the primary results and differences and similarities among studied regions. Meanwhile, mass absorption cross section (MAC) values of BC and WSOC from aerosols, glaciers, snow cover and precipitation, and carbon isotopic signatures ($\Delta^{14}C$, $\delta^{13}C$) from glacier snowpits, precipitation and aerosols are also provided. The data availability and access are provided in Section 5, and the conclusions are summarized in Section 6.

## 2 Research site descriptions

### 2.1 Overview of site distributions

The TP can be divided into three distinct sub-regions, respectively associated with the dominant influence of the westerlies (northern TP), Indian monsoon (southern TP), and the transition region in between, which don't have an exact boundary of each sub-region (Yao et al., 2013). These different features in sub-regions motivated the need for network observations to understand the atmospheric pollutants and their possible impact on the environment. The APCC program currently consists of 29 stations across the TP and its surroundings covering these three distinct sub-regions. Among these stations, there are 27 stations for aerosol sampling and observations, 20 glaciers observed, 3 stations for snow cover observations and 138 locations for snow sampling once, and 6 stations for precipitation observations (Table 1 and Fig. 1). In addition, lake sediment cores were studied from 7 lakes across the Himalayas and TP. According to the distance and the extent of impact from anthropogenic activities, these stations were distributed into two major types, namely urban stations (strongly influenced by anthropogenic activities), and remote stations (weakly impacted by direct anthropogenic emissions). As this paper is the first dataset report based on the APCC program, we will release the carbonaceous dataset from 20 stations for aerosols, 17 glaciers (including samples from surface snow/ice, snowpit and two ice cores), 2 stations for continuous snow cover observations and 138 locations during snow cover surveys, 6 stations for precipitation, and 7 lake sediment cores across the TP and its surroundings. In the future, more comprehensive datasets on mercury, heavy metals, and persistent organic pollutants will also be reported gradually from the APCC program.

### 2.2 Stations for the aerosol and precipitation studies

The APCC network has a total of 29 stations for the aerosol and precipitation studies. In specific, for the spatial distributions (Fig. 1 and Table 1), the APCC program includes 8 stations from Nepal across the central Himalayas to Tibet (Lumbini, Kathmandu, Jomsom, Pokhara, Dhunche, Nyalam, Zhongba, Everest), 3 stations in Pakistan (Hunza, Mardan, Karachi), and 2 stations in the southeast TP (Lulang, Yulong), all of which are dominantly influenced by the Indian monsoon. There are 12 sites (Dushanbe, Toskent, Bishkek, Jimunai, Koktokay, Tianshan, Koxkar, Muztagh Ata, Ngari, Laohugou, Qilian, Beiluhe) distributed in Central Asia, Xinjiang Uygur Autonomous Region (China) and the western and northern TP, mainly controlled by the westerlies. In the inland TP, 2 stations (Nam Co, Tanggula) are distributed across the Nainqentanglha Mountains to the

Tanggula Mountains, which are alternately influenced by the Indian monsoon and westerlies (as the transition sub-region, Yao et al., 2013). Aerosol samples were also collected from Lhasa and Lanzhou city. The elevations of these stations ranged from 13 to over 5000 m a.s.l. (Table 1), with landscapes including forest, alpine steppe, alpine meadow, and desert. These sites served as the key locations for field observations and measurements.

There are 5 stations located on the southern side of the Himalayas in Nepal, focusing on aerosol studies to resolve the transboundary transport of air pollutants (Tripathee et al., 2017; Chen et al., 2020). A list of the detailed information on observation items is in Table 1. Kathmandu, the capital city of Nepal, is characterized by rapid but uncontrolled urban growth and has severe air pollution problems. Pokhara, a famous tourist city, has undergone rapid urbanization with increased numbers of vehicles and industries. Lumbini, located on the northern edge of the Indo-Gangetic Plain (IGP), is a typical rural site located in a mixed setting with a large number of agricultural and industrial activities. Dhunche is a small town situated in the Langtang National Park in the Rusuwa district in the foothills of the Himalayan Mountains; it is approximately 50 km north of Kathmandu. Jomsom is a semi-arid small town in the Mustang district, located in the Kali Gandaki River Valley across the Himalayas.

In Pakistan, 3 stations (namely Karachi, Mardan, and Hunza) are studied based on the APCC program (Gul et al., 2018). Karachi is the capital city of Sindh province, with almost half of the domestic industries in this city. The sampling place is in the extreme south portion, just a few kilometres away from the Arabian sea. Mardan, located near Peshawar city, is the second largest city in Khyber Pakhtunkhwa province. The sampling locations are 10 to 15 km away towards the north side of central Mardan city. Hunza is situated in a valley of Gilgit-Baltistan, on the northern edge of Pakistan, sharing borders with the Wakhan Corridor of Afghanistan and the Xinjiang province of China. The sampling location is at the terminus of Passu glacier and very near to Gulkin and Balthoro glaciers.

In the Central Asian countries, 3 stations were selected and set up for the aerosol studies. Dushanbe is the capital city of Tajikistan. The aerosol sample collection was performed at the Physical Technical Institute of the Academy of Sciences of Tajikistan, which is located in an urban environment on a hill in the eastern part of Dushanbe. Toshkent Shahri is the capital city of Uzbekistan, located in the east part of the country and between the Tianshan Mountains and the Syr river. Bishkek is the capital of Kyrgyzstan, at the foot of the Alatao mountains in north of the country and in the central Chu River basin. All of these sites are located in the arid and semi-arid regions with a dry continental climate.

There are 18 stations (Lanzhou, Lhasa, Everest, Zhongba, Nyalam, Lulang, Yulong, Nam Co, Tanggula, Beiluhe, Qilian, Laohugou, Ngari, Muztagh Ata, Koxkar, Tianshan, Koktokay and Jimunai) continuously observed over the TP and its surroundings within China. Lanzhou is the capital city of Gansu province and is an important industrial base and comprehensive transportation hub in northwest China. Lhasa, China's highest altitude city, is located on the banks of the Lhasa River and serves as the capital of the Tibet Autonomous Region. Everest, Zhongba, and Nyalam are located on the northern side of the Himalayas, characterized by agriculture and yak husbandry and dominated by the Indian monsoon. In particular, at the Qomolangma Atmospheric and Environmental Observation and Research Station of CAS on the north slope of Mt. Everest (Everest station), real-time data of equivalent BC concentrations (eBC, measured by the Aethalometer model, AE-33) has been

observed since May of 2015 (Chen et al., 2018). Lulang is located in a sub-valley of the Yarlung Tsangpo Grand Canyon, a corridor for the warm-humid Indian monsoon to penetrate the inner TP. Yulong, the southernmost glaciated mountain in the Eurasian continent, close to Lijiang City (Yunnan province), may be affected by local emissions. Nam Co is a typical pastoral area in the inland TP. Beiluhe is located to the east of Kekexili of inland TP. Laohugou is a remote hinterland site located in the western Qilian Mountains. Ngari is located in the west TP, a typical arid area covered by bare soil or grasslands. Koxkar station is located on the southern slope of Mt. Tomur, which is the highest peak in western Tianshan. Tianshan Glaciological Station (Tianshan station) is located in the upper reaches of the Urumqi river. Jimunai is located in Northern Xinjiang, near the Altai Mountains.

Precipitation samples in this paper were collected from 6 stations (Fig. 1), namely the Upper Heihe river basin (Qilian), Nam Co, Everest, Lulang, Yulong, and Lhasa city (Gao et al., 2021b; Li et al., 2018b; Niu et al., 2019). The Upper Heihe river basin is a typical permafrost basin, located in the Qilian Mountains of the northern TP (Chen et al., 2014). The other 5 stations have been described in the previous paragraphs.

## 2.3 Glaciers

Carbonaceous data from a total of 17 glaciers are provided in this paper (Table 2, Fig. 1). There are 11 glaciers located in the Indian monsoon dominated region. Among them, Baishui glacier No.1 of Yulong Snow Mountain and the other 4 glaciers (namely Demula, Renlongba, Yarlong, Dongga) are located in the southeast TP (Niu et al., 2018a; Zhang et al., 2017a), and East Rongbuk glacier is located in the central Himalayas on the northern slope of Mt. Everest. In the inland TP, 3 glaciers are observed and studied. The Zhadang glacier, with an area of 2.0 km$^2$ and a length of 2.2 km, is located on the north-eastern slope of the Nyainqentanglha mountain range (Li et al., 2018c). The Xiaodongkemadi glacier, with an area of approximately 1.60 km$^2$ and facing southwest, is located at the headwaters of the Dongkemadi River, a tributary at the upper reaches of the Buqu River near the Tanggula Pass in the central TP (Gao et al., 2012; Li et al., 2017b). The Ganglongjiama glacier (also known as Guoqu glacier) is located on the northern slope of Mt. Geladaindong, the summit peak of the Tanggula Mountains (Hu et al., 2020a). Meanwhile, there are 5 glaciers studied in northern Pakistan. The Passu and Gulkin glaciers are located very near the Karakoram highway connecting Pakistan with China. The Barpu and Mear glaciers are located very close to each other and around 3 km away from the residential area of the Hopar and Nagar valleys and Sachin glacier is close to a small city (Astore) (Gul et al., 2018).

There are 6 glaciers monitored in the westerly-dominated regions. The Laohugou glacier No. 12, a typical valley glacier, is located on the north slope of the western Qilian Mountains in the northern TP and covers an area of 21.9 km$^2$ (Li et al., 2019a; Zhang et al., 2017b). The Muztag Ata glacier has an area of ~0.96 km$^2$ and a length of 1.8 km, located in the eastern Pamir Plateau (Yao et al., 2012). The Anglong glacier is located in the headwater region of Indus, covering an area of 1.5 km$^2$ (Chen et al., 2019c). The Koxkar glacier is located on the south slope of Mt. Tomur, the highest peak in western Tianshan, on the border between China and Kyrgyzstan (Zhang et al., 2017c). The Urumqi glacier No.1 is located at the headwater of the Urumqi River in eastern Tianshan, which is surrounded by the Taklimakan Desert to the south, the Gurbantungut Desert to the

north, and the Gobi Desert to the east. This glacier has an area of 1.65 km$^2$ (50% of the basin) (Li et al., 2019b). The Muz Taw glacier is located on the northern slope of the Sawir mountains in Northern Xinjiang, south of the Ertix river in Central Asia (Zhang et al., 2020b).

There are 2 ice cores retrieved from the East Rongbuk glacier (Mt. Everest region) and the Ganglongjiama (Mt. Geladaindong region) glacier, respectively (Jenkins et al., 2016; Kaspari et al., 2011; Zhang et al., 2015) (Fig. 1). In this study, data from the Muztag Ata glacier and the Urumqi glacier No.1 were not provided, which can be referred to the previous studies (Xu et al., 2012; Wang et al., 2015).

## 2.4 Snow cover

Snow cover samples were collected from 2 stations (the Laohugou and Koktokay) (Fig. 1) and 138 snow survey locations once in the TP and the Northern Xinjiang (Zhang et al., 2018a; Zhong et al., 2019, 2021). The Koktokay Snow Station at the headwater of the Irtysh River was selected as the fixed-point site, which is located in the Kayiertesi river basin, the first tributary of the Irtysh river. The Kayiertesi river basin is in the southern Altai Mountains in China, covering 2365 km$^2$. Seasonally frozen soil and permafrost are widely spread in the basin, and the basin is rich in vegetation (39.2% vegetation coverage in 2014) (Zhang et al., 2016). The land surface is generally covered by forest in the shade and semi-shade and by grassland and shrubs on the sunny and semi-sunny slopes (Zhang et al., 2016). The lowest air temperature is below −45 °C in winter. Average annual maximum snow depth exceeds 1 m (Zhang et al., 2014). Floods in the river basin are due to rain-on-snow events, originating from the combination of rapidly melting snow and intense precipitation. There were 27 locations for snow cover sampling distributed in the southern TP, 10 in the central TP, and 10 from one glacial river basin (Laohugou region in the northern TP) (Zhang et al., 2018a). There were 91 surveyed locations distributed across Northern Xinjiang. Among them, 11 sampling locations were selected during the whole snow season, including 5 locations in the southern Altai Mountains, 3 locations on the west side of the Junggar Basin, and 3 locations in the northern Tianshan Mountains (Zhong et al., 2019).

## 2.5 Sediment cores from the lakes

Lake sediment cores were studied from 7 lakes distributed across the Himalayas and TP (Cong et al., 2013; Neupane et al., 2019) (Table 3, Fig. 1). Gosainkunda and Gokyo are located on the southern slope of Nepal Himalayas. Qiangyong and Ranwu lake are located in the southern and southeast TP, respectively. Nam Co, Lingge Co and Tanggula lake are located in the inland TP. Riverine BC inputs to Qiangyong and Ranwu lake were controlled by the surrounding snow cover and glacier meltwater. All the above lakes are located far from areas with large amounts of anthropogenic activities.

# 3 Sampling in the field

## 3.1 Atmospheric aerosol and precipitation sampling

In this study, carbonaceous data (EC, OC, WSOC) from atmospheric aerosols (TSP, total suspended particulate) were collected from 19 stations based on the APCC program, distributed in the TP, Southeast Asia, and Central Asia. Generally, the TSP were collected on pre-combusted (550 ℃, 6 hours) quartz fibre filters (90 mm in diameter, with pore size of 2.2 μm, Whatman) with a TSP cyclone at a flow rate of 100 L/min for 24 hours (urban or rural sites) or 48 hours (remote sites) (Chen et al., 2019b). The TSP sampler was setup on the roof of the observation building to avoid the effects of ground dust (Fig. 2). After sampling, the filters were kept frozen until analysis. TSP samples were collected every 6 days to bypass the "weekend effect", which indicated that weekly cycle of aerosol composite (with low value sin weekend and high values in the weekdays) was usually governed by the anthropogenic emissions (Satheesh et al., 2011). These quartz filters will be used for measurement of EC, OC, and WSOC from the aerosols.

When precipitation occurred, wet deposition samples were collected by an automated precipitation collector for the analysis of BC and WIOC. After the precipitation event, the precipitation amount was also recorded, and the samples were transferred into HDPE bottles (250 mL) and kept frozen until analysis. Samples of WSOC were collected in prebaked aluminium basins (450 ℃, 6 h) that were placed on a 1.5 m high platform (Li et al., 2016b, 2016c, 2018b, 2021; Niu et al., 2019). Due to limitations in the volume of samples collected during small precipitation events, only those with enough sample amount were selected to determine $\Delta^{14}C$ and $\delta^{13}C$.

## 3.2 Glaciers, snow cover and ice core sampling

Surface snow/ice and snowpits were sampled from glaciers, and meteorological variables in glacial regions were observed by Automatic Weather Stations (Fig. 2). Generally, for glacier surface snow/ice sampling, different types of snow/ice (fresh snow, aged snow and granular ice) samples were collected across the ablation and accumulation zone of the entire glacier (Fig. 3). Whirl-pak bags were used to collect surface snow samples from the upper 0–10 cm (or 0−5 cm) of depth (approximately 2 L, unmelted), as well as surface granular ice samples. In general, the snowpit samples were collected from the accumulation zone of glaciers using a stainless-steel spoon with a vertical depth interval of 5, 10 or 15 cm and transferred into a Whirl-pak bag, following the protocol described by Kang et al. (2007). Generally, duplicate samples were collected to estimate the differences between sampling. *In situ* observations included snow thickness, density, grain size, and surface albedo, which are used to estimate the post-deposition processes, chemicals flux, and effect of BC on glacier melt.

In field surveys across the northern, eastern, southern TP and Northern Xinjiang (Zhang et al., 2018a; Zhong et al., 2019), snow depth, snow density, snow grain size and surface albedo were observed, and snow cover samples were collected. At each location, snow samples were collected from the top 5 cm of the snowpack and stored in a Whirl-pak bag. The vertical resolution of the snowpit profile at Koktokay snow station was 5 cm intervals from the snow surface to the depth of 20 cm, then samples were collected every 10 cm from the depth of 20 cm to the bottom. During the snow accumulation period (November) and the

stable period (from December through early March), snowpit samples were collected three times a month at 10:00 am (UTC/GMT +8.00, the same below). During the intense snowmelt period (from early or mid-March to the early April), snow

samples were collected twice a day at 10:00 am and 7:00 pm, respectively. Snow cover samples for analyses of WSOC were directly collected into the square polycarbonate bottles (Zhang et al., 2019). The samples were kept frozen until they were melted and filtered in the laboratory.

In this paper, historical rBC records from two ice cores were reported. In 2002, a 108 m ice core was collected from the col of the East Rongbuk glacier located on the northeast ridge of Mt. Everest on the northern slope of the Himalayas (Kaspari

et al., 2011). In 2016, a shallow ice core (8 m length) was drilled by the team from the same glacier to expand rBC records since 2000. In November 2005, a 147 m ice core was collected from the upper basin of the Ganglongjiama glacier (Guoqu glacier) on the northern slope of the Mt. Geladaindong using an electro-mechanical drill (Jenkins et al., 2016; Zhang et al., 2015). The drilled ice cores were packed in polyethylene tubing in the field, transported frozen to the State Key Laboratory of Cryospheric Science, Chinese Academy of Sciences in Lanzhou, and kept in a cold room at –20 ℃ until sample preparation

and analysis (Kang et al., 2015).

### 3.3 Lake sediment core sampling

Lake sediment cores were drilled from the deep basin of the studied lakes during 2008−2017 using a gravity coring system with a 6 cm inner diameter polycarbonate tube (Cong et al, 2013; Neupane et al., 2019). The core sediments were sliced in the field at intervals of 0.5 cm, except for Lingge Co and Ranwu Lake, which were sliced at 1 cm intervals, stored in plastic bags,

and kept frozen until analysis.

## 4 Observations, measurement methods and data descriptions

### 4.1 Real-time atmospheric BC observation using Aethalometer

BC is an important part of atmospheric particulate aerosols, which imposes adverse effects on atmospheric visibility, health, and climate change (Ramanathan and Carmichael, 2008). The eBC is operationally defined as the amount of strongly light-

absorbing carbon with the approximate optical properties of carbon soot that would give the same signal in an optical instrument (e.g., Aethalometer) (Andreae and Gelencser, 2006). At the Everest station, a real-time optical instrument (Aethalometer model AE-33, Magee Scientific Corporation, USA) for measurement of eBC from atmospheric aerosols was operated with an inlet installed at approximately 3 m above the ground level since March in 2015. The airflow rate was operated at 4 L min$^{-1}$. The eBC concentrations can be acquired according to the light absorption and attenuation characteristics from

the seven fixed wavelengths (e.g., 370, 470, 520, 590, 660, 880 and 950 nm) (Chen et al., 2018; Drinovec et al., 2015). In general, the eBC concentration measured at 880 nm is used as the BC concentration in the atmosphere, as the absorption of other types of aerosols (e.g., OC and dust) is greatly reduced at this wavelength (Sandradewi et al., 2008). When calculating eBC concentrations, it is therefore possible to eliminate the "loading effect" with the loading compensation parameter $k$, which

allows extrapolation to zero loading, and the accurate ambient BC concentration is obtained (Drinovec et al., 2015). Previous studies have evaluated the real-time compensation algorithm of dual-spot Aethalometer model AE-33 and indicated that AE-33 agreed well with the post-processed loading effect compensated data obtained using earlier Aethalometer models and other filter based absorption photometer (Chen et al., 2018; Drinovec et al., 2015).

At Everest station, daily and monthly mean eBC concentrations presented a strong seasonal variations during 2015−2019, which showed the highest values in the pre-monsoon season (~923 ng m$^{-3}$ in April) and the lowest values in the monsoon season (~88.5 ng m$^{-3}$ in July) (APCC dataset I-1 and Fig. 4). In the pre-monsoon season, BC from the Indo-Gangetic Plain (IGP) region can be transported and concentrated on the southern slope of the Himalayas by the north-westerly winds in the lower atmosphere, and then further transported across the Himalayas by mountain-valley winds (Cong et al., 2015). Simulations further indicated that the BC aerosols in South Asia could be uplifted and transported to the Mt. Everest region by the southerly winds in the upper atmosphere in the monsoon season (Chen et al., 2018). The results indicated the seasonal cycle of BC was significantly influenced by the atmospheric circulation and combustion intensity in the Mt. Everest region. Meanwhile, there were continuously high concentrations of eBC above 1000 ng m$^{-3}$ during 8–10 June 2015, 19–22 March 2016, 9–30 April 2016, and 11–14 April 2017, indicating that the heavy pollution episodes occurred at Mt. Everest during those days (Chen et al., 2018).

### 4.2 Analysis methods and data of atmospheric aerosol EC and OC

After the TSP sampling, the aerosol EC and OC concentrations were measured by a thermal/optical carbon analyser (Desert Research Institute DRI Model 2001 or Sunset Lab) (Chen et al., 2019b). In detail, for the aerosol samples (TSP quartz filters), a punch of sample (with filter area of 0.5 cm$^2$) was put in a quartz boat inside the analyser and heated stepwise at the different temperature plateaus (IMPROVE_A temperature protocol with an optical reflectance correction for sample charring) (Chow et al., 2007). The IMPROVE_A temperature protocol defined temperature plateaus for thermally derived carbon fractions of 140 ℃ for OC1, 280 ℃ for OC2, 480 ℃ for OC3, and 580 ℃ for OC4 in a helium (He) carrier gas and 580 ℃ for EC1, 740 ℃ for EC2, and 840 ℃ for EC3 in a 98% He/2% oxygen (O$_2$) carrier gas. Each carbon fraction reported to the IMPROVE_A network database consisted of a value and a precision. The aerosol OC and EC by thermal/optical reflectance (TOR) were insensitive to the change in such temperature protocol. Therefore, EC, OC, and total carbon (TC) were calculated from the eight carbon fractions as:

$$OC = OC1 + OC2 + OC3 + OC4 + OP \tag{1}$$

$$EC = EC1 + EC2 + EC3 - OP \tag{2}$$

$$TC = OC + EC \tag{3}$$

Where, OP in the equation represented pyrolyzed OC, which was defined as the carbon evolving between the introduction of oxygen in the helium atmosphere and the return of reflectance to its initial values (the OC/EC split) (Chow et al., 2005). The accuracy of the measurement was ±10% and the detection limit for OC, EC and TC were 0.43, 0.12, and 0.49 μg C cm$^{-2}$, respectively (Chen et al., 2019b).

Approximately 1,000 samples from 19 different sites were analysed in this report (APCC dataset I-1). Annual average TC, EC and OC distributions across the TP usually showed higher values in the urban and rural sites and lower values in the remote sites, which considerably decreased from outside to inland of the TP (Chen et al., 2019b) (Fig. 5). The highest TC, EC and OC values were found over urban areas (e.g., Kathmandu: OC=34.8 μg m$^{-3}$, EC=9.97 μg m$^{-3}$; Mardan: OC=44.7 μg m$^{-3}$, EC=11.7 μg m$^{-3}$; Lanzhou: OC=25.4 μg m$^{-3}$, EC=6.7 μg m$^{-3}$), indicating the impact from increased anthropogenic emissions. The OC and EC concentrations in sites on the edge of TP were much lower than those in cities (e.g., Lulang: OC=4.86 μg m$^{-3}$, EC=0.7 μg m$^{-3}$; Hunza: OC=5.12 μg m$^{-3}$, EC=0.78 μg m$^{-3}$; Laohugou: OC=4.81 μg m$^{-3}$, EC=0.59 μg m$^{-3}$), but higher than those in the inland TP regions (e.g., Ngari: OC=1.82 μg m$^{-3}$, EC=031 μg m$^{-3}$; Nam Co: OC=1.63 μg m$^{-3}$, EC=0.13 μg m$^{-3}$). Meanwhile, aerosol TC, EC and OC concentrations revealed apparent seasonality (Fig. 5). Higher concentrations were observed over the South Asian sites (e.g., Lumbini located in northern IGP, and Kathmandu valley) particularly during the pre-monsoon season, due to regional-scale pollution plumes known as atmospheric brown clouds (Ramanathan et al., 2007b). However, different seasonal variations were observed in the inland to northern TP. For example, at Nam Co station, relatively high OC and EC concentrations occurred during the monsoon and post-monsoon. The regional differences in CA seasonal variations suggested differences in the pollutant sources and transport pathways. In the Central Asia, the OC and EC concentrations demonstrated clear seasonal patterns, with elevated concentrations observed during August to February (Chen et al., 2021). These results of different seasonal variations between southern and northern parts of the TP suggested differences in the patterns of pollutant sources and in distance from the sources between the regions (Chen et al., 2019b). All of these measurements have provided the basis for understanding the spatio-temporal variations of carbonaceous particles over the vast Third Pole region, which was of great importance for scientific communities worldwide. Furthermore, these data sets were critical for further scientific studies in the future on the atmospheric environment across this sentinel region.

**4.3 Mass absorption cross section (MAC) of aerosol EC and WSOC from atmospheric aerosols**

The optical attenuation (ATN) can be calculated based on the transmittance signal with equation (4) (Cheng et al., 2011):

$$\text{ATN} = \ln\left(\frac{T_{final}}{T_{initial}}\right) \tag{4}$$

Where, $T_{initial}$ and $T_{final}$ are the transmittance signal before and after the thermal-optical analysis, respectively. The ATN can be used to determine the absorption coefficient ($b_{abs}$) based on equation (5):

$$b_{abs} = \text{ATN} \times \frac{A}{V} \tag{5}$$

Where, $A$ is the filter area (cm$^2$) and $V$ is the volume of air sampled (m$^3$). Thus the MAC of EC (MAC$_{EC}$, m$^2$ g$^{-1}$) can be calculated as:

$$\text{MAC}_{EC} = \frac{b_{abs}}{EC} = \frac{ATN \times A}{EC \times V} = \frac{ATN}{EC_S} \times 10^2 \tag{6}$$

Where, $EC_S$ (μg C m$^{-2}$) is the filter loading of $EC$ concentrations. As multiple scattering effects occur associated with the filter-based measurement of absorption, they were corrected by a value of 3.6 (Chen et al., 2021) (equation 7):

$$\text{MAC}_{\text{EC}} = \frac{ATN}{EC_S} \times 10^2 \times \frac{1}{3.6} \tag{7}$$

The light absorption spectra of WSOC were measured between wavelengths of 200 nm to 800 nm with precision of 5 nm bandwidth by a UV-Visible Spectrophotometer (SpectraMax M5). The value of $\text{MAC}_{\text{WSOC}}$ can be calculated from following equation (8) (Bosch et al., 2014; Kirillova et al., 2014):

$$\text{MAC}_{WSOC} = \frac{A_b}{C \times L} \times \ln(10) \tag{8}$$

Where, $A_b$ (absorbance) is derived directly from the spectrophotometer, $L$ is the absorbing path length, and $C$ is WSOC concentration.

The $\text{MAC}_{\text{EC}}$ at 632 nm exhibited significant spatial variations, differing by a factor of up to two with a clear increasing trend from the outer to the inland TP (Fig. 6). For the study sites, the annual $\text{MAC}_{\text{EC}}$ ranged from 6.37 to 8.49 $\text{m}^2\,\text{g}^{-1}$. Karachi had the lowest $\text{MAC}_{\text{EC}}$ among the sites. Other sites, including Mardan, and Lanzhou, exhibited similar annual average $\text{MAC}_{\text{BC}}$ of approximately 7.0 $\text{m}^2\,\text{g}^{-1}$ during the sampling period. On the southern side of the TP, sites (e.g., Dhunche, Jomsom, Everest, Zhongba, Nyalam, Lulang) had moderate $\text{MAC}_{\text{EC}}$, which were slightly higher than those in urban sites but lower than in the inland remote regions (e.g., Nam Co, Hunza, Ngari, and Beiluhe). In the northern TP, Laohugou demonstrated slightly lower $\text{MAC}_{\text{EC}}$, comparable to that in urban areas. The lower $\text{MAC}_{\text{EC}}$ values in urban areas were mainly affected by local fresh emissions, while the relatively higher $\text{MAC}_{\text{EC}}$ values in remote sites were mostly attributed to the coating enhancement of aerosols (Chen et al., 2019b).

Aerosol $\text{MAC}_{\text{WSOC}}$ were analysed for 10 sites, including Kathmandu, Lumbini, Karachi, Jomsom, Lulang, Everest, Lhasa, Nam Co, Lanzhou, and Laohugou (Chen et al., 2020; Li et al., 2016b, 2016c; Li et al., 2021). Relatively higher $\text{MAC}_{\text{WSOC}}$ was observed in Lumbini, which had an annual average value of 1.64 $\text{m}^2\,\text{g}^{-1}$. Kathmandu, Lanzhou, and Laohugou had moderate values of approximately 1.30 $\text{m}^2\,\text{g}^{-1}$. Jomsom and Karachi had relatively lower $\text{MAC}_{\text{WSOC}}$ values of 0.97 and 0.87 $\text{m}^2\,\text{g}^{-1}$, respectively. The $\text{MAC}_{\text{WSOC}}$ values were generally higher in urban regions than those at remote sites probably because of the larger primary and anthropogenic contribution for WSOC in urban areas in contrast to remote regions (Chen et al., 2020).

## 4.4 BC and WIOC from glaciers and snow cover

Before filtration, these frozen snowpit, surface snow, and snow cover samples were rapidly melted via a hot water bath (approximately 20 minutes for complete melting) and the meltwater (typically 0.5 to 1 L) was filtered through a pre-dried (in a desiccator, at 550 ℃, for 6 hours) weighted quartz filter (Whatman, with pore size of 2.2 μm) using a vacuum pump (Zhang et al., 2017a, 2018; Zhong et al., 2019, 2021). Samples were filtered twice and the filtration equipment was rinsed with ultra-pure water twice (<18.2 mΩ) in order to prevent particle loss. The estimated total uncertainty in the particle concentrations was <1% (including background counts and random counting errors).

After filtering the meltwater, the quartz filters were dried and weighed gravimetrically, then analysed for BC and WIOC. Generally, the dust loads in the snow/ice samples were greater (approximately 2−3 orders of magnitude) than in the airborne aerosol samples. In order to eliminate the impact of dust, BC and WIOC were measured by a modified IMPROVE_A protocol

by DRI model thermal/optical carbon analyser (Yang et al., 2015; Wang et al., 2012). Specifically, the method was modified such that only one temperature plateau (550 $°C$) was used in the 100% helium atmosphere to reduce the time that the BC was exposed to the catalysing atmosphere. The reported OC concentrations from the snow and ice samples account for only WIOC because most of WSOC was not captured by the filter-based method. The detection limit of the analysis was $0.19 \pm 0.13$ μg TC cm$^{-2}$, and the filter blank was $1.23 \pm 0.38$ μg TC cm$^{-2}$, which was about 1 order of magnitude lower than the measured sample values (Zhang et al., 2017).

The evaluated blank filters for total carbon <1 μg cm$^{-2}$. For the same filter, multiple measurements showed small relative standard deviations (RSD, <10%), indicating that the data points tended to be close to the mean value, an acceptable filtration. The duplicate snow samples demonstrated the similar concentrations of BC and WIOC. The evaluated impact of inorganic carbonates interfering with BC measurements showed that the carbonate acidification and analysis indicated acceptable data quality with a discrepancy <20% (Zhang et al., 2017a).

According to the measurements, the average BC concentrations in glacier snow/ice ranged from several ng g$^{-1}$ to hundreds of ng g$^{-1}$, with marked differences between the glaciers (APCC dataset I-2, and Fig. 7). In particular, BC concentrations in the aged snow/granular ice were usually much higher (1−2 orders of magnitude) than that in fresh snow/snowpit/ice cores (Kang et al., 2020). Concentrations of BC and WIOC were higher in the central and northern TP than in the southern TP. There was large spatial variability of BC and WIOC in snow cover across the TP and Northern Xinjiang. Concentrations of BC and WIOC in snow cover over the TP were 202−17468 ng g$^{-1}$ and 491−13880 ng g$^{-1}$, respectively. The values of BC and WIOC in snow cover across Northern Xinjiang varied from 32 to 8841 ng g$^{-1}$, and 77 to 8568 ng g$^{-1}$, respectively. Greater BC and WIOC concentrations in snow cover appeared in the western areas (west of 83 $°E$) than other areas in Northern Xinjiang (Fig. 8). Vertical variations of monthly mean BC and WIOC concentrations in the snowpit profiles showed that the maximum monthly mean BC and WIOC concentrations generally appeared at the snow surface (302−6271 ng g$^{-1}$ for BC and 780−17877 ng g$^{-1}$ for WIOC) (Fig. 9), suggesting that the magnitude of downward migration of BC and WIOC was much less than the enrichment in surface snow for snow cover.

## 4.5 WSOC from glaciers and snow cover

Before the measurement, the melted snow/ice samples (collected using pre-cleaned 60 mL square-shaped polycarbonate bottles) were filtered through a 0.45 μm (pore size) PTFE membrane filter (Macherey–Nagel). WSOC concentrations were determined using a TOC-5000A analyser (Shimadzu Corp, Kyoto, Japan) (Yan et al., 2016; Zhang et al., 2018b). The detection limit of the instrument, precision and average WSOC concentration of the blanks were 0.015 mg L$^{-1}$, ±5% and $0.025 \pm 0.006$ mg L$^{-1}$, respectively, demonstrating that contamination was minimal during the pre-treatment and analysis processing. WSOC concentrations from glacier samples were lower in the snowpit and fresh snow, but higher in the aged snow or granular ice (Fig. 7). For the benchmark glaciers (Muz Taw glacier and Laohugou glacier No.12), the spatial distribution of WSOC generally decreased with increasing elevation, indicating that more intense melting occurred in the lower elevation ablation

zones exposed to higher concentrations of WSOC (Gao et al., 2020; Hu et al., 2018). The detection limit of the analyser was low at $4\ \mu g\ L^{-1}$, while the precision and average WSOC concentrations of the blanks were $\pm 5\%$ and $4 \pm 2\ \mu g\ L^{-1}$, respectively, demonstrating that contamination during the pre-treatment and analysis processing of these samples was weak (Hu et al., 2021).

The light absorption spectra of the WSOC were measured with a UV-Visible Spectrophotometer (SpectraMax M5, Molecular Devices, USA) between the wavelengths of 200 nm and 800 nm with an interval of 5 nm. Each spectrum was determined relative to that of Milli-Q water. The MAC values for WSOC in glaciers and snow cover samples were calculated by equation (8). The $MAC_{WSOC}$ from snowpits of Laohugou glacier No.12 and Ganglongjiama glacier (Central TP) was $4.71 \pm 3.68\ m^2\ g^{-1}$ and $2.17 \pm 2.13\ m^2\ g^{-1}$, respectively (Hu et al., 2018, 2020). $MAC_{WSOC}$ from snow cover in the Altai mountains

was $0.45 \pm 0.35\ m^2\ g^{-1}$, with higher values in March and April 2017; the fraction of radiative forcing caused by WSOC relative to BC accounted for approximately 10.5%, indicating WSOC was a non-negligible light-absorber in snow of the Altai regions (Zhang et al., 2019). The calculated $MAC_{WSOC}$ from Baishui glacier No.1 at Mt. Yulong was $6.31 \pm 0.34\ m^2\ g^{-1}$ (Niu et al., 2018b). The comparisons of $MAC_{WSOC}$ from glaciers and snow cover revealed significant differences between regions (Fig. 10). Even for the same glacier (the Baishui glacier No.1), $MAC_{WSOC}$ showed large variability from snowpits over different

sampling periods (Niu et al., 2020).

## 4.6 WSOC from precipitations

The precipitation samples were also filtered through a PTFE membrane filter with 0.45 μm pore size (Macherey–Nagel) before the WSOC measurement. The Shimadzu TOC-5000 total organic carbon analyser (Shimadzu Corp, Kyoto, Japan) was used to determine the precipitation WSOC concentration (Li et al., 2016d). The average blanks for precipitation WSOC was $0.08 \pm 0.05$

$\mu g\ C\ mL^{-1}$ (Gao et al., 2021b). The light absorption spectra of precipitation WSOC were measured by using the same method used for the glacier samples. The value of precipitation $MAC_{WSOC}$ was calculated by the equation (8).

    Precipitation WSOC concentrations decreased from urban cities to remote stations, with marked seasonal variations (APCC dataset I-3, and Fig. 11) (Gao et al., 2021b; Li et al., 2017a, 2018a; Niu et al., 2019). The average precipitation WSOC concentrations was $1.41\ \mu g\ C\ mL^{-1}$ at the Qilian Station of the northern TP, with WSOC flux of $6.42\ kg\ ha^{-1}\ yr^{-1}$ (Gao et al.,

2021b). For summer precipitation in the Mt. Yulong region in southeast TP, the average concentration of WSOC was $1.25\ \mu g\ C\ mL^{-1}$ (Niu et al., 2019). The average precipitation WSOC concentration at Nam Co was $1.0\ \mu g\ C\ mL^{-1}$; the estimation suggested that about15% of WSOC was fossil derived (Li et al., 2018a). The $MAC_{WSOC}$ values of precipitation samples were significantly lower than the aerosol samples (Li et al., 2017a). For example, the precipitation $MAC_{WSOC}$ only ranged $0.26$–$1.84$ $m^2\ g^{-1}$ at Qilian Station, suggesting the potential impact of WSOC on climatic forcing in the area. Seasonally, $MAC_{WSOC}$ of

both aerosol and precipitation samples showed high values in winter and low values in summer (Gao et al., 2021b; Li et al., 2021).

## 4.7 Carbon isotopes from glacier snowpits and atmospheric aerosols

The dual-carbon-isotope signatures ($\Delta^{14}C$ and $\delta^{13}C$) are effective ways to distinguish and track the different sources of carbon aerosols (Gustafsson et al., 2009). In this study, BC mass contents of the filtered glacier snowpit and atmospheric aerosols (TSP) samples were quantified using the TOT technique. Briefly, filters were acidified by fumigation in open glass Petri dishes held in a desiccator with >37% HCl acid for 24 hours to remove carbonates and were subsequently dried at 60 °C for 1 hour to remove remaining HCl acid. A 1.5-cm$^2$ punched acid-treated filter was analysed using a carbon analyser (Sunset Laboratory, Tigard, OR) following the National Institute for Occupational Safety and Health (NIOSH) method 5040 to determine BC and OC concentrations of the aerosol samples and the WIOC concentrations of the particles from the collected snow samples. Sucrose standard and other reference materials were also subjected to these measurements (Li et al., 2016). The filter area required for the subsequent $^{14}C$ measurements was determined based on the measured BC concentration. The $CO_2$ produced was cryotrapped during the BC combustion phase after removing the water and sulphur-containing gases online. Purified $CO_2$ was then transferred in flame-sealed glass ampules to the United States National Science Foundation (US-NSF) National Ocean Science Accelerator Mass Spectrometry (NOSAMS) facility at the Woods Hole Oceanographic Institution (Woods Hole, MA, USA). For precipitation and snow samples containing more than 60 µg C (WSOC), the sample was poured into pre-cleaned quartz tubes, acidified to pH 2 with phosphoric acid, and sparged with ultrahigh purity helium to remove inorganic carbon. Next, the sample was irradiated using a high-energy UV lamp for 5 hours to quantitatively oxidize WSOC to $CO_2$. Concentrations of WSOC were determined using a calibrated Baratron absolute pressure gauge (MKS Industries). WSOC concentrations determined by this method well agreed with those determined using the Shimadzu TOC analyser (Raymond et al., 2007). $CO_2$ was cryogenically purified by liquid nitrogen on a vacuum extraction line and sent to the National Ocean Sciences Accelerator Mass Spectrometry (NOSAMS) at Woods Hole for isotopic analysis.

Fossil fuel contribution to BC in both aerosol and snowpit samples decreased from the outer TP (i.e., South Asia and western China) to the inland TP (APCC dataset I-4, and Fig. 12) (Li et al., 2016). For the Himalayan region (Thorung, East Rongbuk, Qiangyong), equal contributions from fossil fuel (46±11%) and biomass (54±11%) combustion were found, consistent with BC source fingerprints from the IGP region (Dasari et al., 2020). BC in the remote northern TP (Laohugou) predominantly derived from fossil fuel combustion (66±16%), consistent with Chinese sources. The fossil fuel contributions to BC in the inland TP (Tanggula, Zhadang) were lower (30±10%), implying contributions from internal Tibetan sources (for example, yak dung combustion). A similar phenomenon was observed for WSOC in snowpits, which revealed lower fossil fuel contributions in the inner TP. The results suggested that pollutants transported from South Asia influenced the BC and WSOC concentrations over the southern part of TP. Meanwhile, Chinese emissions influenced northern TP and local emissions influenced inland TP.

## 4.8 rBC data from ice cores

In the laboratory, the drilled ice core sections were cut longitudinally into halves, which were sectioned using a modified band-saw (stainless-steel blades; table tops and saw guides covered with Teflon) cleaned with ethyl alcohol and ultrapure water (18.2 MΩ). One half-section of the core was cut into 3−5 cm segments. For the East Rongbuk ice core, a total of 663 samples were collected for rBC analysis (Kaspari et al., 2011). For Geladaindong ice core, a total of 3585 samples were collected for rBC analyses (Jenkins et al., 2016). To eliminate possible contamination from sampling, drilling or storage, the outer portion of each ice sample was scraped using a ceramic knife in a clean and low-temperature room (–8 ℃). Polypropylene clean room-suits and non-powder vinyl clean-room gloves were worn throughout the sampling process to minimize potential contamination (Kang et al., 2007). After the outer section of the ice was removed, the samples were put into Whirl-pak bags, melted at room temperature, and the solution was poured into high-density polyethylene vials for the subsequent experiments. The samples were stored as liquid until analysis (Jenkins et al., 2016).

The rBC records from ice cores were analysed by using a SP2 (Droplet Measurement Technologies), an important analytical technique applied to Arctic ice cores (McConnell et al., 2007). The SP2 uses laser induced incandescence to measure the refractory BC mass in individual particles quantitatively and independent of particle morphology and coatings with light scattering material, with detection limit was 0.3 fg/particle (Schwarz et al., 2012). Coupled to a nebulizer, the SP2 can be used to measure rBC particles in liquid-phase samples (Lim et al., 2014; Schwarz et al., 2013). The high sensitivity and small-required sample volume enabled ice cores to be analysed at much higher resolution than thermal-optical methods. Generally, rBC concentrations from ice cores are likely to be systematically underestimated due to the nebulization efficiency. The nebulization efficiency is size dependent, with large (>500 nm) and small (<200−250 nm) BC particles nebulized with lower efficiency than mid-sized particles (Wendl et al., 2014; Schwarz et al., 2012). Besides, ice core samples were stored in the liquid phase after melting, and prior findings indicate that storage in the liquid phase can result in as great as 80% reduction in measured BC concentrations (Wendl et al., 2014; Kaspari et al., 2014). Nevertheless, because we focus on the temporal variation of rBC from ice core for the historical records rather than on absolute concentrations, the systematic underestimation of rBC does not affect the historical changes.

Historical records from the East Rongbuk ice core showed that rBC concentrations have increased approximately threefold from 1975–2000 relative to 1860–1975 (APCC dataset I-5, and Fig. 13), inferring anthropogenic BC is transported to high elevation regions of the Himalaya (Kaspari et al., 2011). The Geladaindong ice core provided the first long-term rBC records in the central TP spanning 1843−1982; after the 1940s, the rBC record was also characterized by an increasing trend (Fig. 13). Such an increase in rBC concentrations over the recent decades was likely attributed to the increased combustion emissions from regional BC sources, and a reduction in snow accumulation (Jenkins et al., 2016).

**4.9 BC data from lake sediment cores**

The freeze-dried lake sediment core samples were grinded into powder with a size <0.074 mm using an agate mortar, and approximately 0.10−0.15 g of sample was transferred into a 50 mL centrifuge tube. In order to remove carbonates, silicates and metal oxides, the sediment samples were acid treated with 10 mL of HCl (2 N) and left for 24 hours at room temperature. The supernatants were removed and rinsed with ultrapure water. Then, 15 mL of the mixture of HCl (6 N) and HF (48% v/v) with ratio of 1:2 (v/v) was added to the residue for further digestion for 24 h at room temperature and subsequently rinsed thoroughly with ultrapure water. The residue was treated with HCl (4 N) at 60 ℃ overnight to get rid of fluoride that may have formed, which was then centrifuged to remove the supernatant liquid. The residue was rinsed with pure water until the pH of the eluent became neutral. Finally, the residual solid was diluted with 200 mL of ultrapure water and filtered through a 47 mm quartz fibre filter ensuring even distribution on the filter surface, which was then analysed for BC abundances (Cong et al., 2013).

Generally, the IMPROVE-A protocol with TOR has been used for determination of BC concentrations in lake sediments (Neupane et al., 2019). The repeated measurement of BC in a few lake sediments (n=5) were to ensure reproducibility of measurements, which reported as relative percentage deviation was better than 8%. Standard reference material (marine sediments, NIST SRM-1941b) were also analysed to assess the accuracy of measurements. It indicated an average accuracy of 5.5% for the measurements of BC in lake sediments in this study.

The BC flux ($g\ m^{-2}\ yr^{-1}$) of the lake sediment cores was calculated by multiplying the BC concentration ($mg\ g^{-1}$) with the quotient of total weight of sample (dry weight per layer) and area of gravity corer (6 cm inner diameter polycarbonate tube; area = $\pi \times r^2$), as well as the deposition time (i.e., the time difference between the two consecutive deposition layers):

$$\text{BC flux} = \frac{BC\ concentration \times \frac{Dry\ wt.per\ layer}{Area\ of\ gravity\ corer}}{Deposition\ time} \qquad (9)$$

Historical BC records from the lake sediment cores show an increasing trend of BC concurrent with increased anthropogenic emissions since the 1950s (APCC dataset I-5, and Fig. 14). The relatively constant trend of BC before the 1950s could be attributed to the background level with minimal inputs from anthropogenic activities (Cong et al., 2013; Neupane et al., 2019). Previous studies pointed out that the deposition of BC in lake sediment cores was mainly related to river transport from the lake basin as a result of climate change (e.g., increases in temperature and precipitation) (Li et al., 2017). The much higher BC flux in recent decades may also be caused by increases in direct atmospheric deposition in addition to riverine input.

# 5 Data availability

All the BC, OC, WIOC, WSOC, MAC values of BC and WSOC, and BC isotope datasets presented in this paper have been released and are available to download from the National Cryosphere Desert Data Center, Chinese Academy of Sciences (http://www.ncdc.ac.cn) at Lanzhou (Kang S et al., 2021. Black carbon and organic carbon dataset over the Third Pole. National Cryosphere Desert Data Center, 2021. https://doi.org/10.12072/ncdc.NIEER.db0114.2021).

A specific directory was designated with data classified into different categories:

a) Aerosol BC and OC abundances, and their MAC data (APCC dataset I-1),

b) Glacier and snow cover BC, WIOC, and WSOC data, and MAC values of WSOC (APCC dataset I-2),

c) Carbon isotope data from snowpits and aerosol (APCC dataset I-3),

d) Precipitation WSOC and BC data and MAC values of WSOC (APCC dataset I-4), and

e) BC data from ice cores and lake sediment cores (APCC dataset I-5).

In each dataset, a short summary is also provided. Auxiliary data including site descriptions (e.g., locations), observation
contents, and measurements are presented.

## 6 Conclusions

The dataset of BC and OC concentrations and their related MAC values and carbon isotope signatures from the atmosphere, glaciers, snow cover, precipitation, and lake sediment cores based on the APCC program over the Third Pole are presented in this paper. These data are a collaborative effort aimed to address multiple scientific issues, including: (1)
characterizing the carbonaceous components and depicting their spatial and temporal variations over the TP and its surrounding region; (2) identifying the carbon aerosol sources and investigating the mechanisms of long-range transport; and (3) constraining the role of carbon components on the glacier/snow cover melting. Our continuous efforts based on the current APCC program are the in-depth investigations of the origins and distributions of atmospheric pollutants and their impacts and response to cryospheric changes over the Third Pole region. Moreover, this paper presents the long-term spatial analysis of
carbonaceous particles from multi environmental samples (atmosphere, glacier, lake sediments, etc.) over the Third Pole. The data from this work are also significant for understanding the negative consequences of pollution on environment of remote regions and pave a way for future perspectives and protection strategies.

## Competing interests

The authors declare that they have no conflict of interests.

**Acknowledgements**

We would appreciate all our international collaborators (Prof. Örjan Gustafsson, Prof. Chhatra Mani Sharma, Dr. Maheswar Rupakheti, Dr. Arnico Panday), engineers (Dr. Xiaoxiang Wang, Dr. Shaopeng Gao, Mr. Yajun Liu), and others (Dr. Shiwei Sun, Dr. Bin Qu, Zhengzheng Yang, Chengde Yang, Huijun Zheng), who participated in the field observations, maintained the observation instruments, processed the observations, and analysed the samples in the laboratory.

## Financial support

This work was supported by the Second Tibetan Plateau Scientific Expedition and Research program (2019QZKK0605), the Strategic Priority Research Program of Chinese Academy of Sciences (XDA19070501, XDA20040500), Stake Key Laboratory of Cryospheric Science (SKLCS-ZZ-2022), the National Natural Science Foundation of China (42071082), and CAS "Light of the West China" Program.

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

**Table 1** Detailed geographic characteristics of the Atmospheric Pollution and Cryospheric Change program observation stations across the Tibetan Plateau and its surrounding in this paper.

| Regions | Abbreviations | Site | Latitude (°N) | Longitude (°E) | Elevation (m a.s.l.) | Observations |
|---------|---------------|------|---------------|----------------|----------------------|--------------|
| Altai | JMN | Altai Observation and Research Station of Cryospheric Science and Sustainable Development (Jimunai, North Xinjiang) (roof height 2m) | 46.843 | 88.133 | 997 | Muz Taw glacier, aerosol |
| | KKTH | Koktokay Snow Station, North Xinjiang (roof height 2m) | 47.353 | 89.662 | 1379 | snow cover |
| Tianshan | TS | Tianshan Glaciological Station, Xinjiang (roof height 2m) | 43.105 | 86.807 | 2100 | Glacier No.1 at Urumqi River source region, aerosol, snowpit |
| | KQK | Koxkar (Tianshan), Xinjiang (roof height 2m) | 41.813 | 80.17 | 3000 | Koxkar glacier, aerosol |
| Tibetan Plateau | MT | Muztagh Ata Station for Westerly Environment Observation and Research, Western Tibetan Plateau (roof height 2m) | 38.291 | 75.055 | 5725 | Muztagh Ata glacier, aerosol |
| | NGR | Ngari Station for Desert Environment Observation and Research, Western Tibetan Plateau (roof height 2m) | 33.392 | 79.701 | 4270 | Anglong glacier, aerosol |
| | LHG | Qilian Observation and research Station of Cryosphere and Ecologic Environment, Northern Tibetan Plateau (roof height 2m) | 39.429 | 96.556 | 4230 | Laohugou glacier No.12, snow cover, aerosol |
| | QL | Qilian Alpine Ecology and Hydrology Research Station, Northern Tibetan Plateau | 38.25 | 99.8667 | 3040 | precipitation |
| | BLH | Beiluhe Observation and Research Station on Frozen Soil Engineering and Environment in Qinghai-Tibet Plateau (roof height 2m) | 35.428 | 92.556 | 4000 | aerosol |
| | TGL | Tanggula Cryosphere and  Environment Observation Station, Central Tibetan Plateau | 33.083 | 92.067 | 5000 | Xiandongkemadi/Ganglongjiama glacier |
| | NMC | Nam Co Station for Multisphere Observation and Research, Southern Tibetan Plateau (roof height 2m) | 30.779 | 90.991 | 4730 | Zhadang glacier, snow cover, aerosol, precipitation |
| | ZB | Zhongba, Southern Tibetan Plateau (roof height 2m) | 29.7 | 83.983 | 4704 | aerosol |
| | NLM | Nyalam, Southern Tibetan Plateau (roof height 2m) | 28.167 | 85.983 | 4166 | aerosol |
| | EV | Qomolangma Atmospheric and Environmental Observation and Research Station (Everest), Southern Tibetan Plateau (roof height 2m) | 28.35 | 86.933 | 4276 | East Rongbuk glacier, aerosol, precipitation |
| | SETS | South-East Tibetan plateau Station for integrated observation and research of alpine environment (Lulang), Southeast Tibetan Plateau (roof height 2m) | 29.767 | 94.733 | 3326 | aerosol, precipitation, Demula/Yarlong/Renlongba/Dongga glacier |
| | YL | Yulong Snow Mountain Glacial and Environmental Observation and Research Station, Southeast Tibetan Plateau (roof height 2m) | 27.167 | 100.167 | 2650 | Baishui glacier No.1, aerosol, precipitation |
| Related cities | LZH | Lanzhou city, Gansu Province (roof height 25m) | 36.05 | 103.859 | 1520 | aerosol |
| | LS | Lhasa city, Xizang Province (roof height 15m) | 29.633 | 91.3 | 3642 | aerosol, precipitation |
| Nepal | DC | Dhunche (roof height 3m) | 28.117 | 85.3 | 2051 | aerosol |
| | PKR | Pokhara (roof height 6m) | 28.183 | 83.983 | 813 | aerosol |
| | JMS | Jomsom  (roof height 3m) | 28.767 | 83.717 | 3048 | aerosol |
| | KTMD | Kathmandu (roof height 15m) | 27.683 | 85.4 | 1300 | aerosol |
| | LMB | Lumbini  (roof height 15m) | 27.483 | 83.283 | 100 | aerosol |
| Pakistan | KRC | Karachi (roof height 10m) | 24.85 | 66.983 | 13 | aerosol |
| | HZ | Hunza (roof height 3m) | 36.46 | 74.892 | 2519 | Passu/Gulkin/Barpu/Mear/Sachin glaciers, snow cover, , aerosol |
| | MD | Mardan (roof height 10 m) | 34.239 | 72.048 | 485 | aerosol |
| Central Asia | DSB | Dushanbe (roof height 3m) | 38.5588 | 68.8558 | 864 | aerosol |
| | TSK | Toshkent (roof height 10m) | 41.2667 | 69.2167 | 821 | aerosol |
| | BSK | Bishkek (roof height 2m) | 42.8833 | 74.7666 | 750 | aerosol |

**Table 2** Detailed information for the observed glaciers based on the Atmospheric Pollution and Cryospheric Change program across the Third Pole.

| Regions | Mountains | Glacier name | Latitude | Longitude |
|---|---|---|---|---|
| Southeastern Tibetan Plateau | Hengduanshan | Baishui glacier No.1 | 27.17 ꞌN | 100.15 ꞌE |
| Southeastern Tibetan Plateau | Nyainqengtanglha Mts. | Demula glacier/Renlongba glacier/Yarlong glacier/Dongga glacier | 29.355 ꞌN | 97.02 ꞌE |
| Himalayas | Mt. Everest, Himalayas | East Rongbuk glacier | 28.031 ꞌN | 86.961 ꞌE |
| Inland Tibetan Plateau | Nyainqengtanglha Mts. | Zhadang glacier | 30.467 ꞌN | 90.633 ꞌE |
| Inland Tibetan Plateau | Tanggulha Mts. | Xiaodongkemadi glacier | 33.067 ꞌN | 92.067 ꞌE |
| Inland Tibetan Plateau | Tanggulha Mts. | Gangklongjiama glacier (Guoqu glacier) | 33.833 ꞌN | 91.683 ꞌE |
| Northern Tibetan Plateau | Qilian Mts. | Laohugou glacier No.12 | 39.44 ꞌN | 96.542 ꞌE |
| Western Tibetan Plateau | Mt. Anglonggangri, Ngari | Anglong glacier | 32.849 ꞌN | 80.932 ꞌE |
| Tianshan | Wester Tianshan | Keqikaer glacier | 41.813 ꞌN | 80.17 ꞌE |
| Tianshan | Eastern Tianshan | Urumqi glacrei No.1 | | |
| Northern Xinjiang | Sawir Mts. | Muz Taw glacier | 47.06 ꞌN | 85.56 ꞌE |
| Eastern Pamir Plateau | Mt. Muztagh | Muztagh Ata glacier | 38.283 ꞌN | 75.067 ꞌE |
| Northern Pakistan | Karakoram and Himalayas | Passu glacier | 36.45 ꞌN | 74.85 ꞌE |
| Northern Pakistan | Karakoram and Himalayas | Gulkin glacier | 36.42 ꞌN | 74.77 ꞌE |
| Northern Pakistan | Karakoram and Himalayas | Barpu glacier | 36.18 ꞌN | 74.08 ꞌE |
| Northern Pakistan | Karakoram and Himalayas | Mear glacier | 36.15 ꞌN | 74.82 ꞌE |
| Northern Pakistan | Karakoram and Himalayas | Sachin glacier | 35.32 ꞌN | 74.76 ꞌE |

Note: An ice core with depth of 108 m was collected from the col of the East Rongbuk glacier (28.03 ꞌN, 86.96 ºE, 6518 m) located on the northeast ridge of Mt. Everest; An ice core with depth of 147 m was collected from the upper basin of the Ganglongjiama glacier (Guoqu glacier, 33.58 ꞌN, 91.18 ꞌE; 5750 m a.s.l.) on the northern slope of the Mt. Geladaindong. Mts.: Mountains

**Table 3** Detailed information for the lake sediment cores across the Third Pole in this paper.

| Regions | Lake name | Latitude | Longitude | Elevation (m a.s.l.) |
|---|---|---|---|---|
| Tibetan Plateau | Ranwu Lake | 29.441 °N | 96.796 °E | 3800 |
| | Qiangyong Co | 28.89 °N | 90.226 °E | 4866 |
| | Nam Co | 30.779 °N | 90.991 °E | 4730 |
| | Tanggula | 32.903 °N | 91.953 °E | 5152 |
| | Lingg Co | 33.831 °N | 88.603 °E | 5051 |
| Nepal | Gokyo | 27.951 °N | 86.69 °E | 4750 |
| | Gosainkunda | 28.095 °N | 85.65 °E | 4390 |


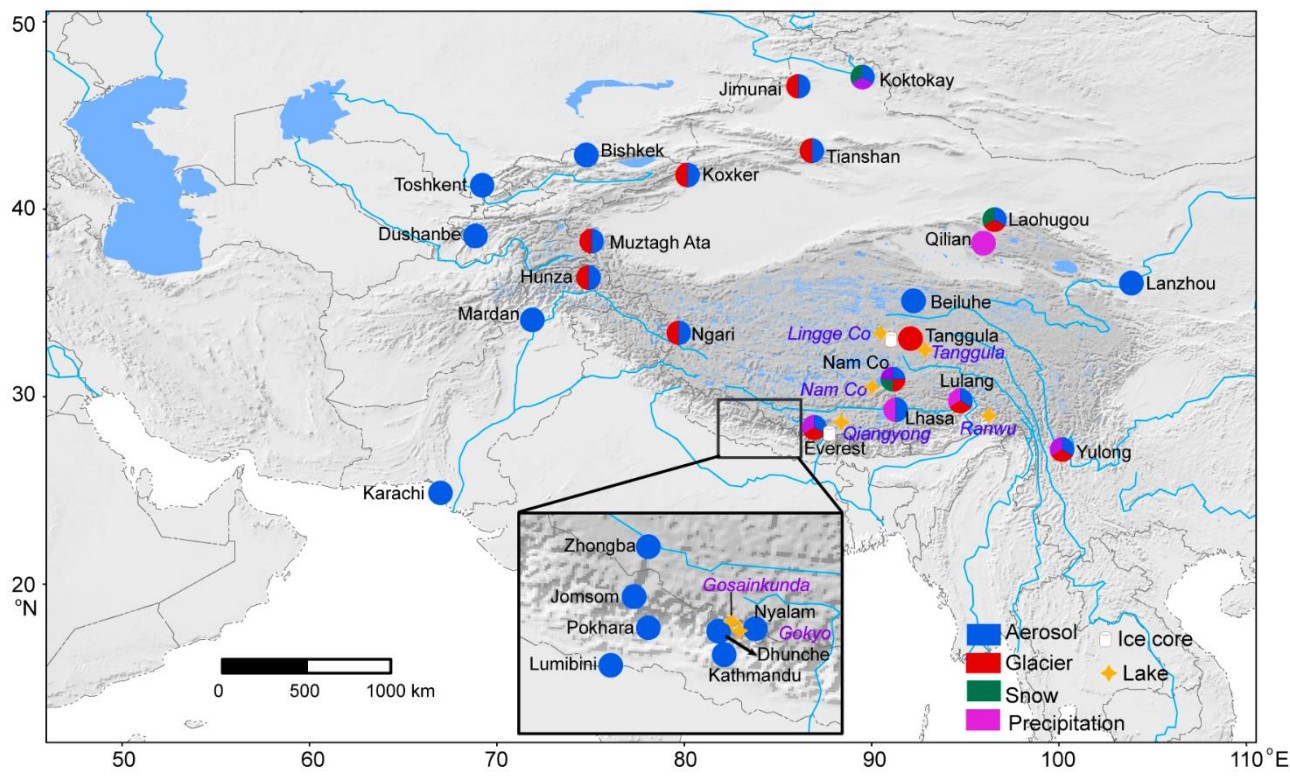

**Figure 1.** Schematic overview of measurements from the Atmospheric Pollution and Cryospheric Changes program over the Third Pole, in which coordinated carbonaceous component measurements were made on samples from the atmosphere, glaciers, snow cover, precipitation, and lake sediment cores. The location information for each station, glacier and lake are provided in Tables 1, 2 and 3.

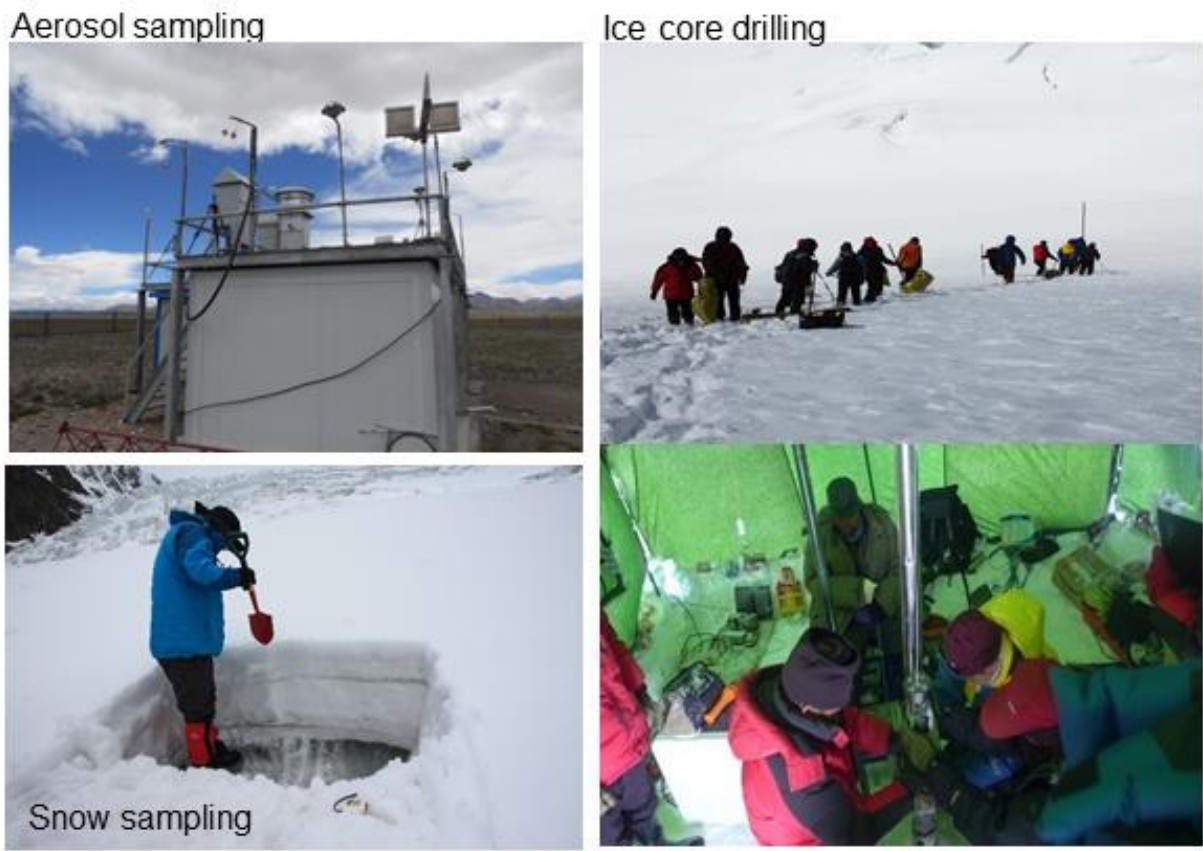

**Figure 2.** Photos of aerosol sampling, snow sampling, and ice core drilling on the Tibetan Plateau.

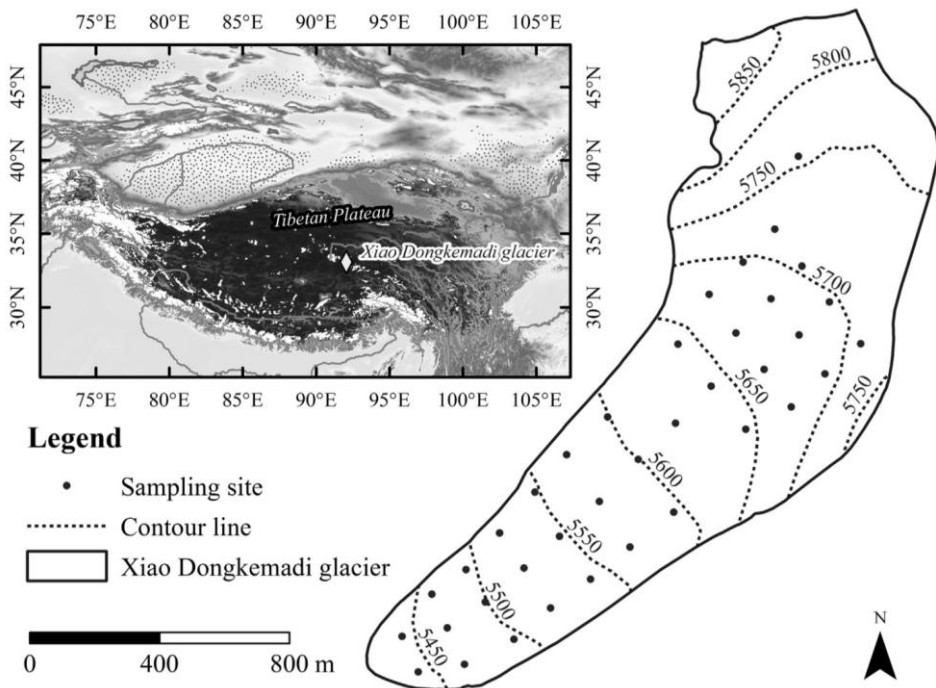

**Figure 3.** Black carbon and organic carbon data in surface snow and snowpit sampling sites from the Xiaodongkemadi glacier at Tanggula
Mountains in the central Tibetan Plateau. Modified from (Li et al., 2017a).

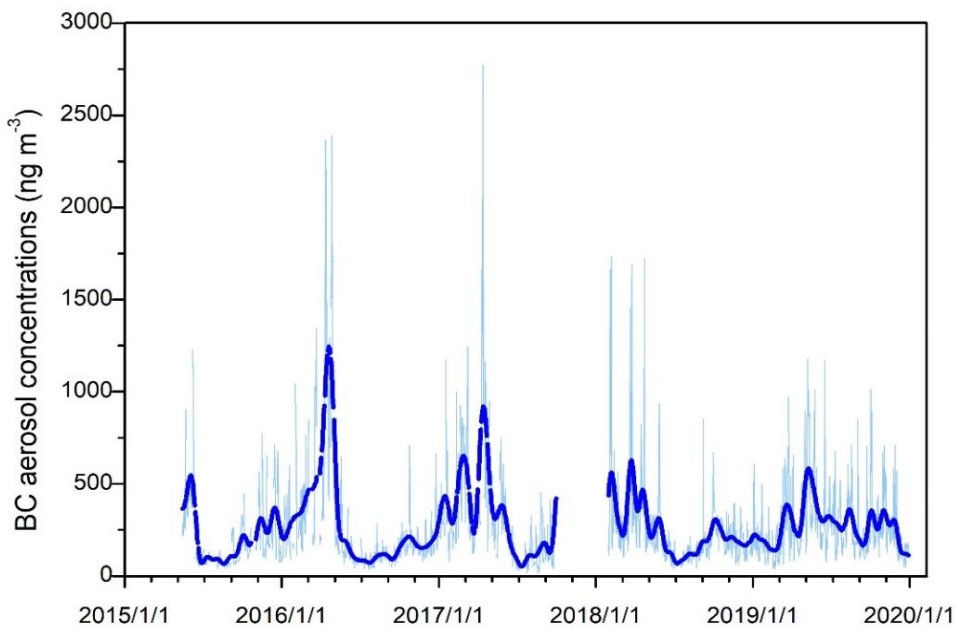

**Figure 4.** Daily mean BC aerosol concentrations at Everest Station (Mt. Everest region) measured by AE-33 during May15 in 2015 to December 29 in 2019. Light blue lines refer to daily data; the thick blue lines represent the smoothing results.


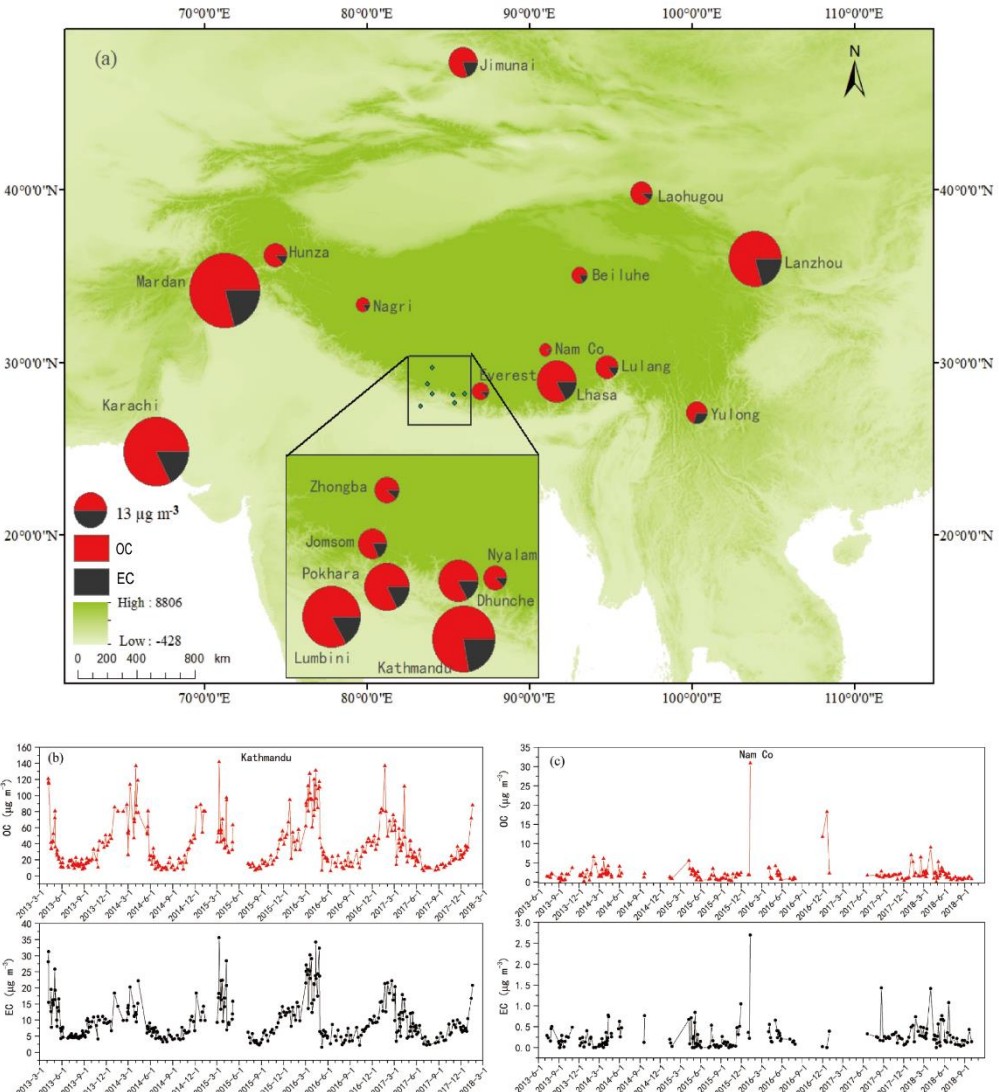

**Figure 5.** Aerosol OC and EC concentration distributions in the Tibetan Plateau and its surroundings (top), and temporal variations of OC and EC concentrations at Kathmandu and Nam Co station (bottom) during the observed periods, respectively. The abbreviations can be found in Table 1.


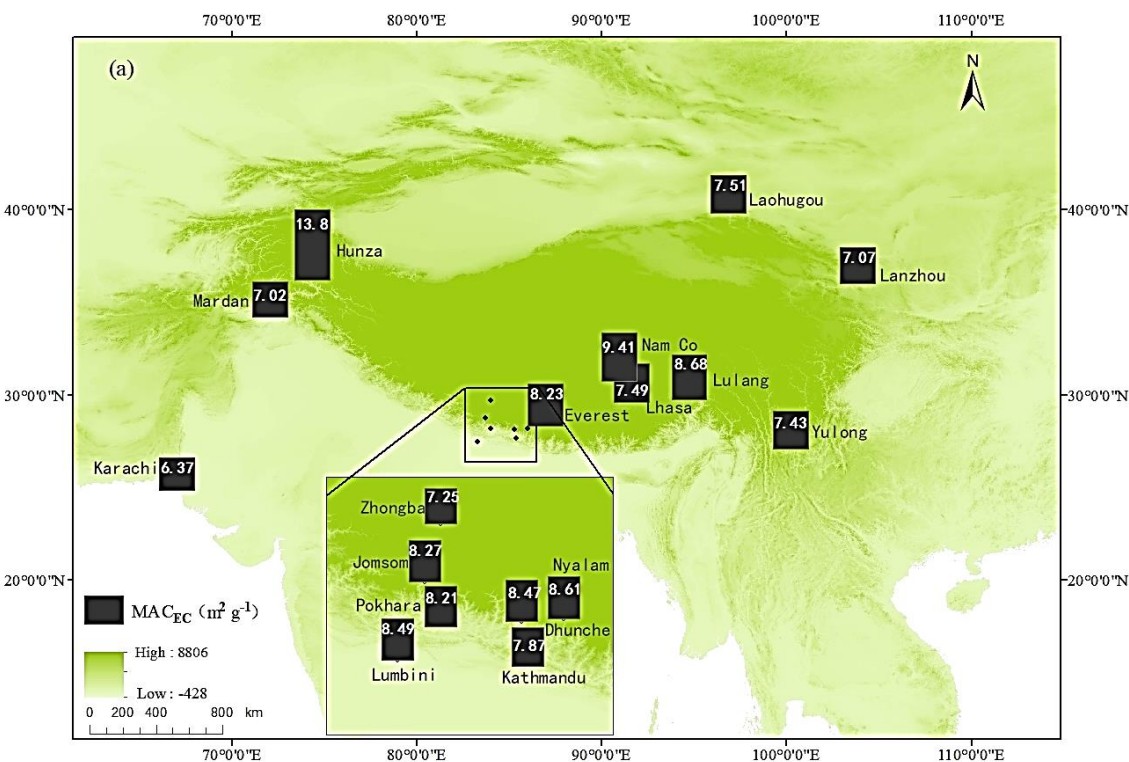

**Figure 6.** Spatial distribution of mass absorption cross section of EC (MAC$_{EC}$) (annual average value) on the Tibetan Plateau. Modified from (Chen et al., 2019b).

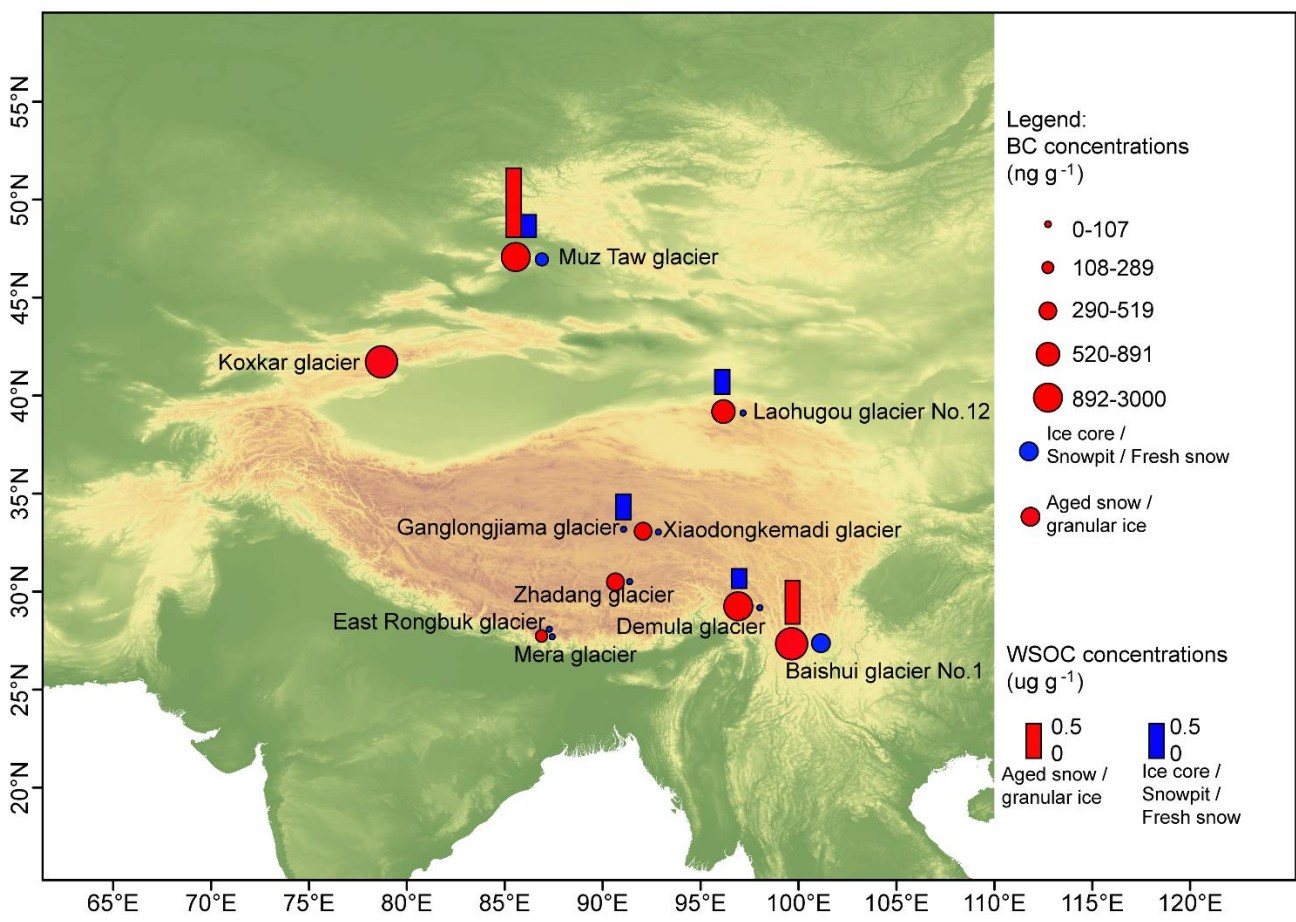

**Figure 7.** BC and WSOC concentrations from studied glaciers distributed on the Tibetan Plateau and surroundings.

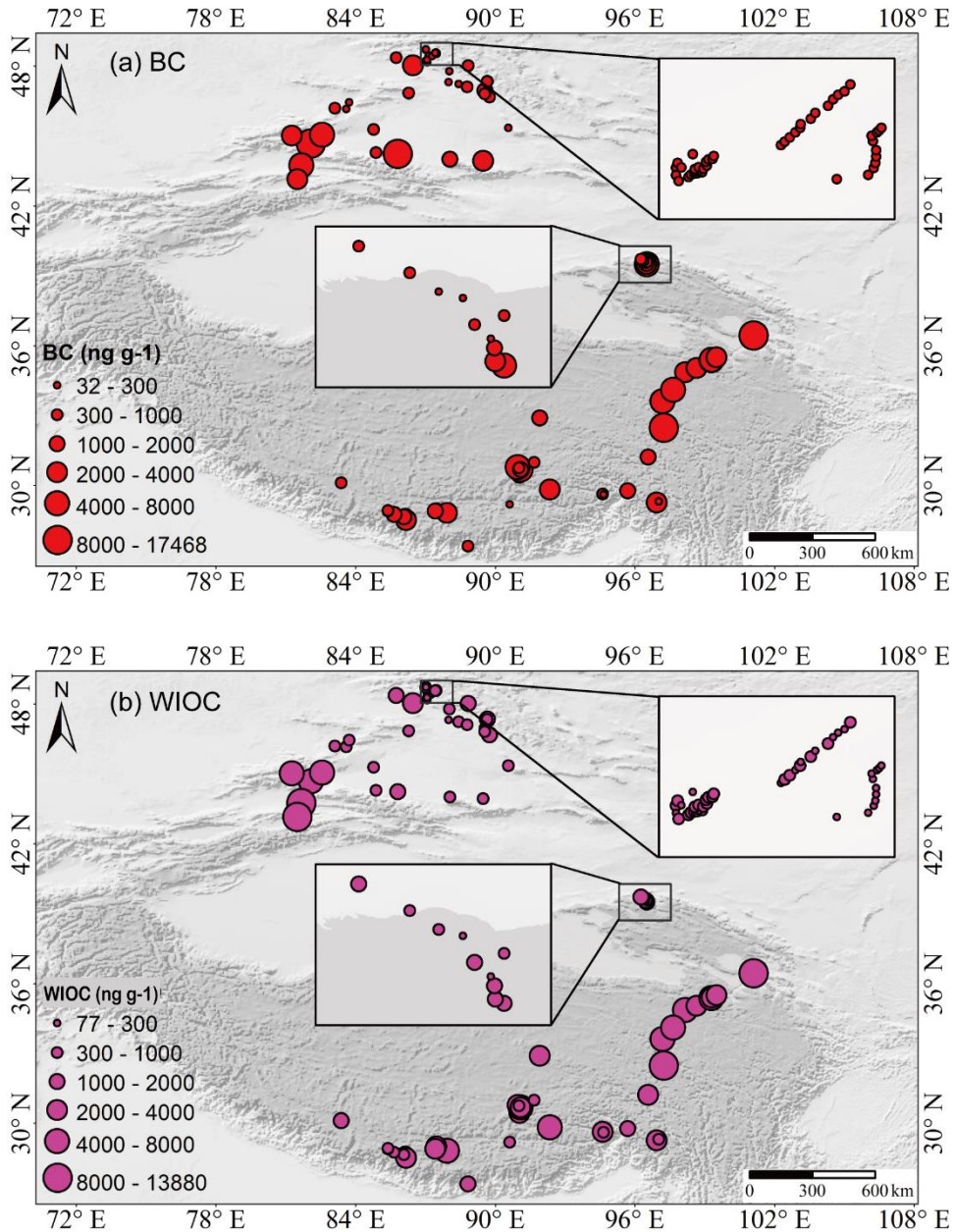

**Figure 8.** Spatial distributions of (a) BC and (b) WIOC concentrations in snow cover for each sampling site across the Tibetan Plateau and the Northern Xinjiang. Data cited from (Zhang et al., 2018a; Zhong et al., 2019).


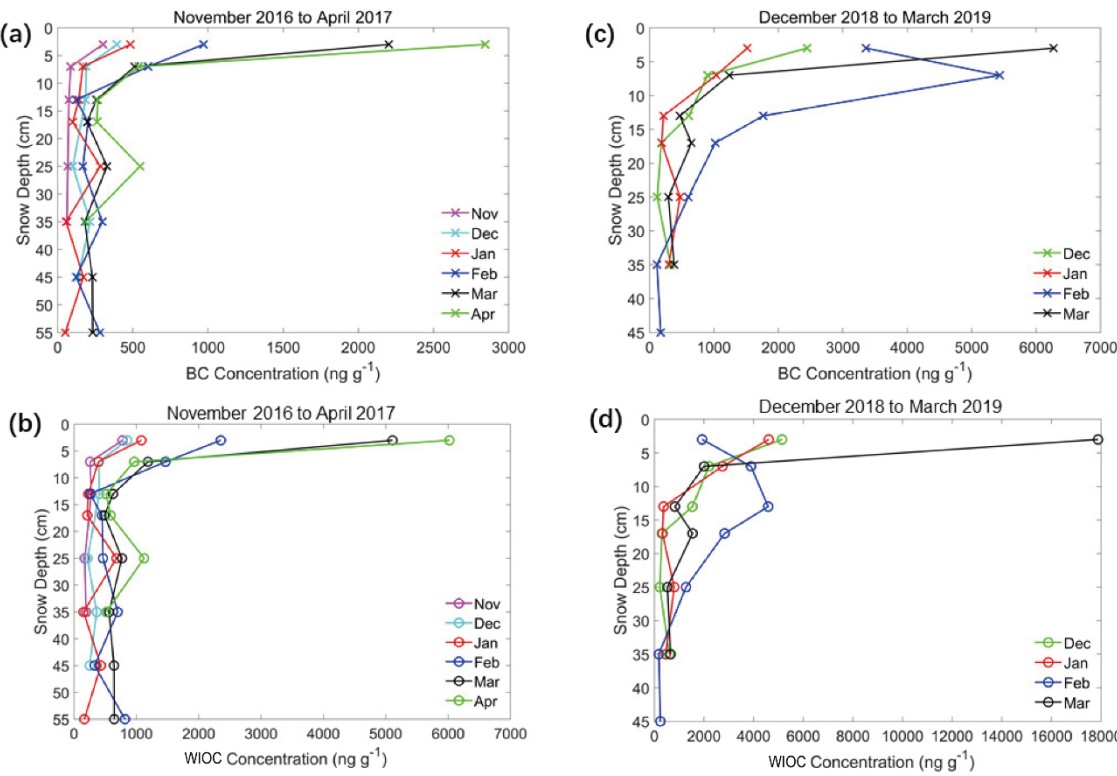

**Figure 9.** Vertical variations of mean BC and WIOC concentrations in snowpits at Koktokay snow station during two snow years (Zhong et al., 2019).

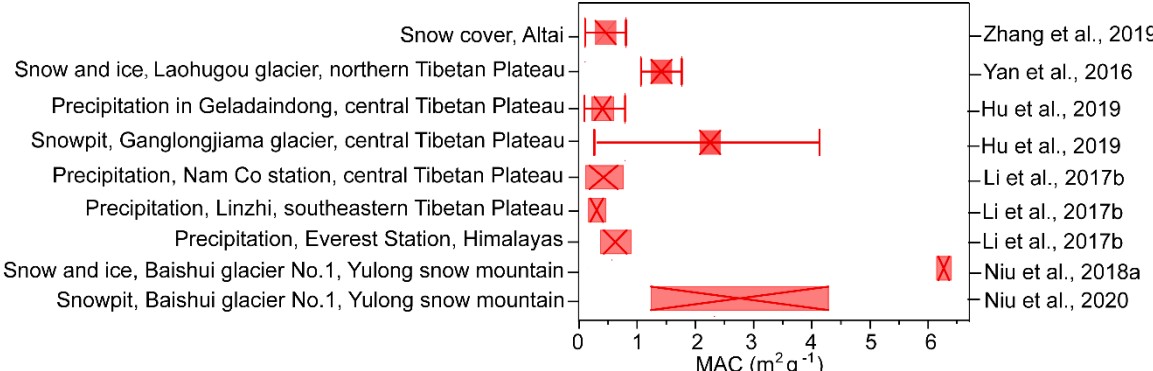

**Figure 10.** Comparison of the mass absorption cross section (MAC) of WSOC in snow and ice, and precipitation across the Tibetan Plateau and its surroundings.

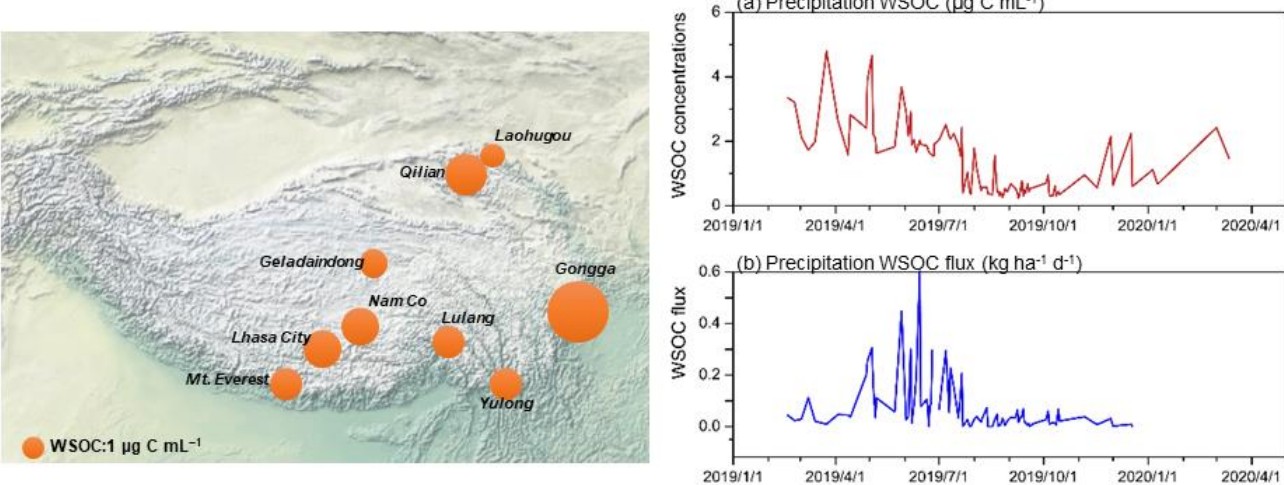

**Figure 11.** Spatial distributions of precipitation WSOC on the Tibetan Plateau and its surroundings (left), and WSOC temporal variations of concentrations and flux from precipitation at Qilian Station in the northern Tibetan Plateau (right). (a) Precipitation WSOC concentrations and (b) Precipitation WSOC flux (modified from Gao et al., 2021b).

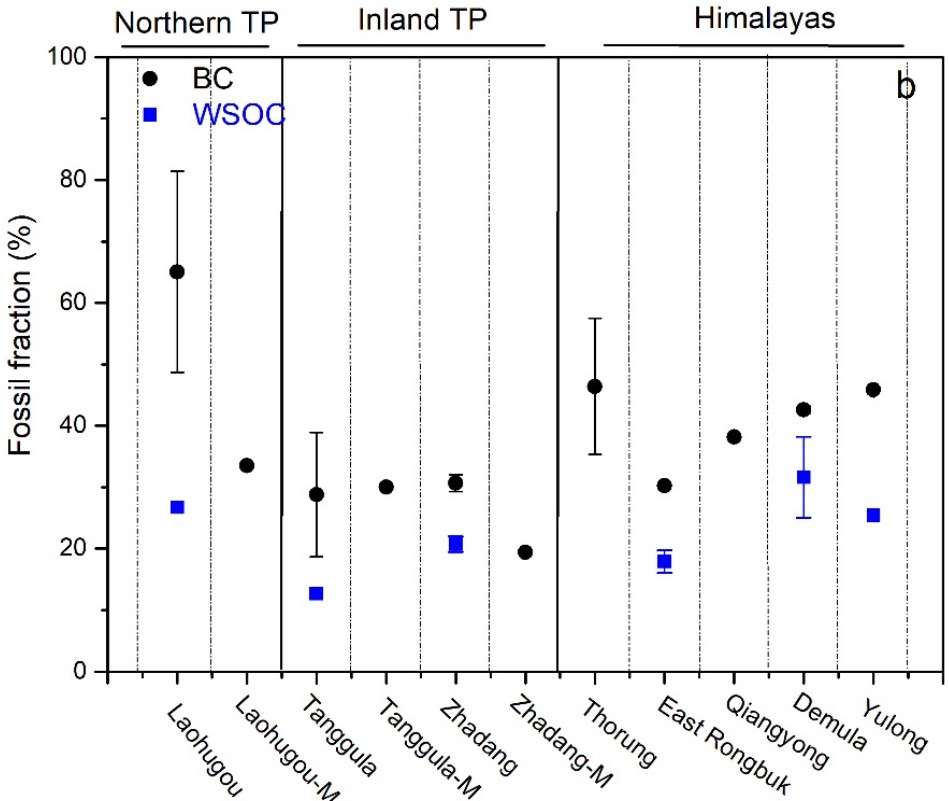

**Figure 12.** BC carbon and WSOC isotopes from snowpit samples in the Himalayas and Tibetan Plateau. Modified from (Li et al., 2016a).

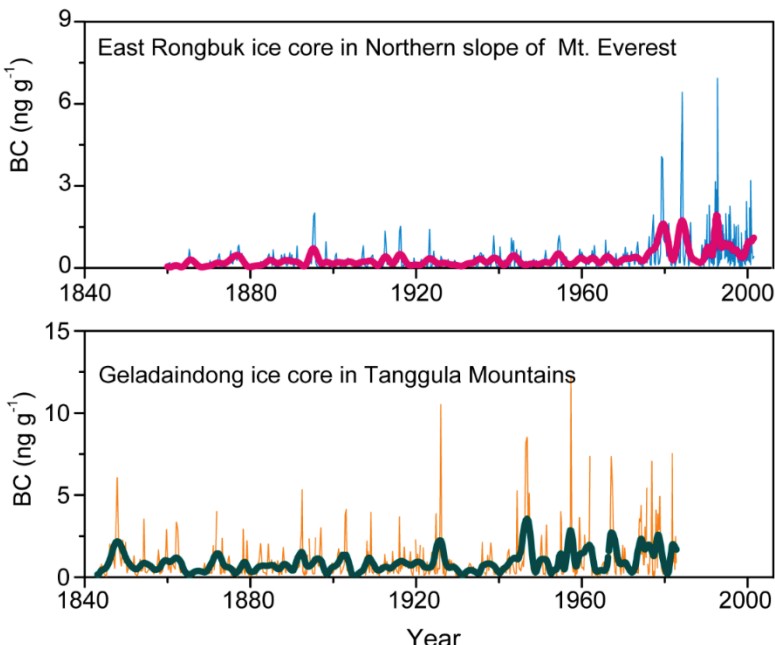

**Figure 13.** Historical BC concentration records retrieved from ice cores over the Himalayas and Tibetan Plateau. East Rongbuk ice core data from the northern slope of Mt. Everest in Himalayas (Kaspari et al., 2011), and Geladaindong ice core data from the Tanggula Mountains of the central Tibetan Plateau (Jenkins et al., 2016).

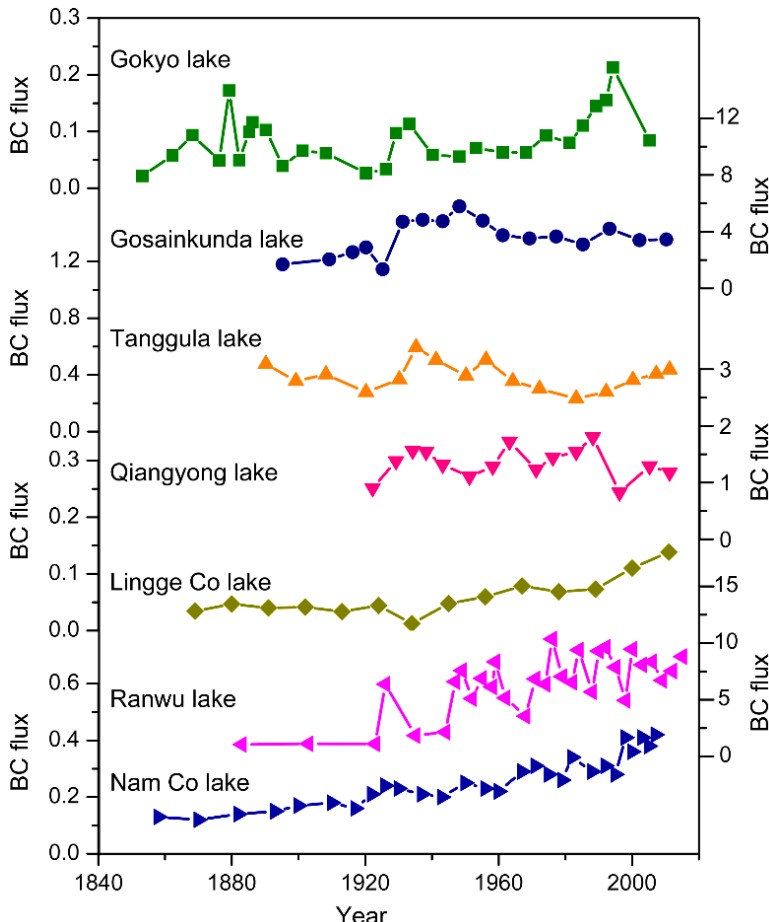

**Figure 14.** Historical BC flux (g m$^{-2}$ yr$^{-1}$) retrieved from lake sediment cores across the Third Pole. Nam Co lake sediment core data was cited from (Cong et al., 2013); other lake sediment cores data was cited from (Neupane et al., 2019).