# Peer review of "Black carbon and organic carbon dataset over the Third Pole"

_Earth System Science Data, 2021_

## Author Comment (AC1)

Nov 08, 2021

Dear Editor and the reviewer,

Thank all of you very much for taking the time and efforts to review our manuscript, titled with "APCC Data Report I: Black carbon and organic carbon dataset from atmosphere, glaciers, snow cover, precipitation, and lake sediment cores over the Third Pole", that we have submitted to "Earth System Science Data" (MS No.: essd-2021-187). We have considered all comments and suggestions carefully and tried our best to address them and revised the manuscript accordingly. We greatly appreciate all the constructive comments and suggestions that have led to an improvement of the paper, and we hope the revised manuscript is suitable for publication in the journal.

Revisions are made to address the following specific comments.

Our point-by-point responses to the comments are listed below in blue color.

Yours sincerely,

Shichang Kang and Yulan Zhang on behalf of all co-authors

**Response to comments:**

Review on paper #ESSD-2021-187: The manuscript entitled "APCC Data Report I: Black carbon and organic carbon dataset from atmosphere, glaciers, snow cover, precipitation, and lake sediment cores over the Third Pole " by Kang and colleagues present a systematic data report on black carbon and organic carbon from various environmental medias including atmosphere, glaciers, snow cover, precipitation, and lake sediment cores over the Third Pole. The authors setup an observation network named atmospheric pollution and cryospheric change (APCC) covering the Tibetan Plateau (TP) and its surrounded region, noting their efforts for APCC are very commendable. The data reported in the manuscript from the APCC is valuable and crucial for understanding the atmospheric pollution and their impact on cryosphere on TP, where continually observations are scares. The topic of the paper is of great importance, and within the scope of the journal Earth System Science Data. The manuscript is logically organized, well structured, nicely written, and the figures and tables are appropriate. I suggest a minor revision, and recommend the authors address the following suggestions before

publication.

Answer: Thank you very much for all these comments.

Line 112: This paper reported data from APCC. However, the description about the network is too short. Please provide more detail information, eg. the background of the network setup, the beginning and development of the network.

Answer: APCC network namely the "Atmospheric Pollution and Cryospheric Changes" has been thoroughly introduced by Kang et al. (2019). In this paper, we have provided more information on the APCC. As shown, in 2013, we initiated a coordinated APCC monitoring network with the overarching goal of performing more integrated and in-depth investigations of the origins and distributions of atmospheric pollutants and their impacts on cryospheric changes over the Third Pole region. Meanwhile, the specific goals of APCC network are listed as flowing:

(I) Characterize the chemical compositions and levels of atmospheric pollutants and depict their spatial and seasonal variation over the Third Pole region;

(II) Identify the source regions of atmospheric pollutants and reveal the pathways and mechanisms by which atmospheric pollution is trans-boundary transported to the Third Pole region;

(III) Investigate the role of atmospheric pollutants deposited as Light Absorbing Impurities (LAPs) in the melting of glacier ice and snow cover and, further, quantify the contribution of LAIs to glacier and snowpack melting, and determine the fates of environmentally relevant pollutants within glaciers and snowpack and their scavenging processes during the melting of snow and ice.

Certainly, as development, we have also modified our goals according to the research progress. Currently, we also focused the feedbacks of cryospheric melting on carbon cycle and hydrology in the Third Pole (Gao et al., 2021). The research area is not only focused on the Third Pole, but also extended to the central Asia (Chen et al., 2021). Besides, new emergent pollutants (for example, microplastics) has been observed and sampled (Zhang et al., 2021). The monitoring prototype, observational and sampled process, measurements and quality control were all

introduced (Kang et al., 2019).

Therefore, in this study, we briefly provided the main structure and observations in the main text. According to the suggestion, we have also tried to added the related new information, which is not redundant from our already published paper (Kang et al., 2019)

References:

Chen, P., Kang, S., Abdullaev, S. F., Safarov, M. S., and Li, C.: Significant influence of carbonates on determining organic carbon and black carbon: a case study in Tajikistan, central Asia. Environ. Sci. Technol. 55, 5, 2839–2846, doi: 10.1021/acs.est.0c05876, 2021.

Gao, T., Zhang, Y., Kang, S., Abbott, B.W., Wang, X., Zhang, T., Yi, S., Gustafsson, Ö.: Accelerating permafrost collapse on the eastern Tibetan Plateau. Environ. Res. Lett., doi: 10.1088/1748-9326/abf7f0, 2021.

Kang, S., Zhang, Q., Qian, Y., Ji, Z., Li, C., Cong, Z., Zhang, Y., Guo, J., Du, W., Huang, J., You, Q., Panday, A.K., Rupakheti, M., Chen, D., Gustafsson, Ö., Thiemens, M.H., and Qin, D.: Linking Atmospheric Pollution to Cryospheric Change in the Third Pole Region: Current Progresses and Future Prospects. Nat. Sci. Rev., 6, 4, 796−809, doi: 10.1093/nsr/nwz031, 2019.

Zhang, Y., Kang, S., Gao, T., Kang, S., Shangguan, D., Luo, X.: Albedo reduction as an important driver for glacier melting in Tibetan Plateau and its surrounding areas. Earth Sci. Rev., 220, 103735, doi: 10.1016/j.earscirev.2021.103735, 2021a.

Zhang, Y., Kang, S., Wei, D., Luo, X., Wang, Z., Gao, T.: Sink or source? Methane and carbon dioxide emissions from cryoconite holds, subglacial sediments, and proglacial river runoff during intensive glacier melting on the Tibetan Plateau. Fundamental Research,1, 232-239, doi: 10.1016/j.fmre.2021.04.005, 2021b.

Zhang, Y., Gao, T., Kang, S., Allen, S., Luo, X., Allen, D.: Microplastics in glaciers of the Tibetan Plateau: evidence for long-range transport of microplastics. Sci. Total Environ., 758, 143634, doi: 10.1016/j.scitotenv.2020.143634, 2021c.

Line 126-127: Do these three domains have exact bound? Please add the longitude and latitude if have.

Answer: The three domain don't have the exact boundary. In Yao et al. (2013)'s study, they

only show the schematic boundaries in grey lines separating the three domains (the westerlies domain, the transition domain, and the monsoon domain) (Fig. R1). The boundary lines are not in strait lines. It is difficult to add the exact longitude and latitude for each domain.

[Figure]

Figure R1. Schematic boundaries of three domains over the Tibetan Plateau. (Yao et al., 2013)

References:

Yao, T., Masson-Delmotte, V., Gao, J., Yu, W., Yang, X., Risi, C., Sturm, C., Werner, M., Zhao, H., He, Y., Ren, W., Tian, L., Shi, C., and Hou, S.: A review of climatic controls on δ18O in precipitation over the Tibetan Plateau: Observations and simulations. Rev. Geophys., 51, 4, 525–548, doi: 10.1002/rog.20023, 2013.

Line 142 and 145: sometimes you use '2 stations' but sometimes 'two stations'. Please use consistent expression.

Answer: Agree, we have revised to keep them in the consistent expression.

Line 166-171: you introduced some aerosol sampling sites in central Asia but didn't provide OC and EC data of these sites.

Answer: The sites in central Asia were included in our APCC network. However, BC and OC data from these sites were not analyzed until we submitted our manuscript to the journal. Currently, Chen et al. (2021) have reported the variations of OC and EC from aerosol in Central

Asia. And we have added the related data in this study now.

References:

Chen, P., Kang, S., Abdullaev, S. F., Safarov, M. S., and Li, C.: Significant influence of carbonates on determining organic carbon and black carbon: a case study in Tajikistan, central Asia. Environ. Sci. Technol. 55, 5, 2839–2846, doi: 10.1021/acs.est.0c05876, 2021.

Line 178: why only detected rBC in the site? Why chose Mt. Everest station?

Answer: Do you mean eBC here? We only have one equipment to measure the eBC, which has been setup at the Mt. Everest station to investigate the trans-boundary transport of BC aerosols. Therefore, we can only provide the eBC data from this station currently. Meanwhile, Mt. Everest region

Line 195, 200: again use three, five glaciers. Please check the whole manuscript.

Answer: Sure, we have checked through the entire manuscript. At the beginning of the related sentences, we use the six or five, not the numbers.

Line 200: "Five glaciers studied in the Karakoram…" confused about this sentence.

Answer: Here we mean we have observed 5 glaciers in the northern Pakistan regions (part of Karakoram and western Himalayan region).

Line 240: why do you collected TSP but not PM2.5? I think fine particles are easily to transport to remote regions. Actually, I think APCC will be of great importance to scientific communities worldwide. However, by now, it seems that the APCC only observes BC, EC, OC and some other related indexes. Will you observe other important atmospheric pollutant in the future? for example, PM2.5, POPs.

Answer: Based on APCC network, we have truly studied the other pollutants, for example, mercury, PAHs, and microplastics (Huang et al., 2019; Zhang et al., 2021; Zheng et al., 2020). But in this article, we focused to report the carbonaceous aerosols rather than other chemicals. As we mentioned in the abstract, in the future, datasets of mercury, heavy metals, and POPs will be reported.

In this study, OC and EC are retrieved from TSP samples. PM2.5 samples were not collected at most of the station due to the harsh environment (limited power, cold and high-elevations). Therefore, we reported the data analyzed from TSP samples.

References:

Huang, J., Kang, S., Ma, M., Guo, J., Cong, Z., Dong, Z., Yin, R., Xu, J., Tripathee, L., Ram, K., Wang, F.: Accumulation of atmospheric mercury in glacier cryoconite over western China. Environ. Sci. Technol., 53, 6632-6639, 2019.

Zhang, Y., Gao, T., Kang, S., Allen, S., Luo, X., Allen, D.: Microplastics in glaciers of the Tibetan Plateau: evidence for long-range transport of microplastics. Sci. Total Environ., 758, 143634, doi: 10.1016/j.scitotenv.2020.143634, 2021c.

Zheng, H., Kang, S., Chen, P., Li, Q., Tripathee, L., Maharjan, L., Guo, J., Zhang, Q.: Sources and spatio-temporal distribution of aerosol polycyclic aromatic hydrocarbons throughout the Tibetan Plateau. Environ. Pollut., 261, 114144, doi: 10.1016/j.envpol.2020.114144, 2020.

Line 243: Could you give more description on the roof the TSP sampler setup? For example, how high the roof is? Is it different for the remote sites and urban sites? Do you think that the height of the roof influences the sampling?

Answer: The detailed information about height of roof where the TSP sampler setup has added in Table 1 in the main text. In urban and rural sites, most samples are set on the roof of building with height of 10 or 15m. In remote sites, most are about 2 or 3 meters, which are based on each station' infrastructure. The height can affect the sampling if set near the surface. Therefore, we set all sampler height higher than 2 m.

Line 265, 266: does the 'snowpit' and 'snowpack' represent the same meaning? If yes, use one for easy reading.

Answer: Snowpit is used for studied glaciers, and snowpack were used for snow cover.

Line 290: the eBC was measured for TSP? Why it is BC but not EC? Because for aerosol

samples, you use EC in section 4.2.

Answer: The eBC represented the equivalent black carbon, the abbreviation is generally used as eBC. The eBC was measured on-line, not off-line analysis from the TSP samples. For BC measured in TSP samples, it is also name as EC which equals to BC.

Line 302: The title of 4.2 is not clear. What is Atmospheric aerosol EC and OC methods?

Answer: The sub-section title has been revised as *__Analysis methods and data of atmospheric aerosol EC and OC__*.

Line 327: please check the manuscript, some words were deleted using revision mode.

Answer: We have revised. Thank you.

Line 365: this has been described in section 3, thus the first sentence can be deleted here.

Answer: Agree, and deleted.

Line 367: what is the pore diameter of the quartz filter? Please clarify because if the pore size is big, some particles will be lost.

Answer: the pore size of quartz filters is 2.2 μm. In Li et al. (2016)' study, we have estimated the efficiency of filtration. It was reported that the ratio of the BC contents in samples with and without NH4H2PO4 was determined to be 77±17%, which denotes a fairly high degree of recovery.

References:

Li, C., Bosch, C., Kang, S., Andersson, A., Chen, P., Zhang, Q., Cong, Z., Chen, B., Qin, D., Gustafsson, Ö.: Sources of black carbon to the Himalayan-Tibetan Plateau glaciers. Nat. Comms., 7, 124574, doi: 10.1038/ncomms12574, 2016.

Line 390, 411: again there are some words deleted using revision mode.

Answer: We have revised.

Line 394: the author introduced the blanks information for WSOC measurement, but didn't

provide information for other equipment.

Answer: We have tried to added the measurements accuracy, limits, and blanks for each equipment mentioned in the study.

Line 507: put 'and' before 'samples'

Answer: Done.

Table 1: it seems the abbreviations are not used in this manuscript. In addition, can you add the location of every site in TP (eg. southern, northern). This would be useful for reading.

Answer: The abbreviations is used in the Data tables. Thus, we provided the abbreviations here. The location information of latitude, longitude, and elevations are also provided.

Figure 1: is that possible to add the boundary of three domains?

Answer: As we mentioned, there is no exact boundary of each domain, thus we didn't put the boundary in the figure.

Caption of figure 3: please use correct reference style.

Answer: Revised.

Figure 10: put the long descriptions in the title but not in the figure.

Answer: The descriptions of sample types and locations are for each reference, which make them clear to understand.

Figure 11: add unit to the title of y-axis.

Answer: Added, and thank you for all of these comments.

---

## Author Comment (AC2)

Nov 08, 2021

Dear Editor and the reviewer,

Thank all of you very much for taking the time and efforts to review our manuscript, titled with "APCC Data Report I: Black carbon and organic carbon dataset from atmosphere, glaciers, snow cover, precipitation, and lake sediment cores over the Third Pole", that we have submitted to "Earth System Science Data" (MS No.: essd-2021-187). We have considered all comments and suggestions carefully and tried our best to address them and revised the manuscript accordingly. We greatly appreciate all the constructive comments and suggestions that have led to an improvement of the paper, and we hope the revised manuscript is suitable for publication in the journal.

Revisions are made to address the following specific comments.

Our point-by-point responses to the comments are listed below in blue color.

Yours sincerely,

Shichang Kang and Yulan Zhang on behalf of all co-authors
* * *
**Response to comments:**

The paper "APCC Data Report I: Black carbon and organic carbon dataset from atmosphere, glaciers, snow cover, precipitation, and lake sediment cores over the Third Pole" by Kang et al., presents a comprehensive summary of data report regarding the field measurements of black carbon and organic carbon in the Tibetan Plateau, known as the Third Pole. Generally, the paper is clear-written, and the dataset is important to the community for investigating the effect of anthropogenic impacts on the carbon cycle and the potential climate feedback in high elevation regions. I only have a few comments before publications:

Answer: Thank you very much for all the structured comments to improve our manuscript.

- There is a lack of descriptions about the data quality control and what methodology has been used for data processing.

Answer: The data quality for each equipment or parameter reported in this study has been added. The BC and OC data provided in our study are based on the different samples, the processes of

sample collection are according to the standard protocols used. For example, in study by Zhang et al. (2017), for BC and POC in snow samples, we evaluated blank filters for total carbon <1 $\mu g\ cm^{-2}$. For the same filter, multiple measurements showed small relative standard deviations (RSD, <10%), indicating that the data points tended to be close to the mean value, an acceptable filtration (Fig. R1). To test if the filtered meltwater volume contributed to BC measurement error, we filtered the carbon mass concentrations using different volumes. Corresponding relative standard deviations were mostly <30% (Table 1). This implied that our filtration procedure was reasonable. The duplicate snow samples demonstrated the similar concentrations of BC and OC (Fig. R2). We also evaluated the impact of inorganic carbonates interfering with BC measurements at the Laohugou No. 12 glacier, we acidified (N37% HCl) selected filters by fumigation in open glass Petri dishes held in a desiccator for 24 h and subsequently dried at 60 °C for 1 h to remove any remaining HCl. The results from the carbonate acidification and analysis indicated acceptable data quality with a discrepancy <20% (Fig. R3).

[Figure]

Figure R1 Multiple measurements of (a) BC concentrations and (b) ratios of OC to BC for

selected samples. (Red square represents the mean value.)

[Figure]

Figure R2 (a) BC and (b) OC concentrations for duplicate snow samples (collected in Aug 2015) filtered with different volumes. (A-K mean the sampling sites on the Laohugou No.12 glacier as shown in Fig. 1b.)

[Figure]

Figure R3 Comparison of BC concentrations measured with and without acidification.

**Table 1**
BC and OC concentrations (ng g$^{-1}$) for snow samples collected in August 2015 and the corresponding RSDs (%) filtered with different volumes.

[revised manuscript text omitted]

- The manuscript lacks discussions about the scientific aspects of this dataset. I understand this is a data description paper but the manuscript in the current form looks more like a technical report, rather than a data paper. I notice there are several plots adapted from previous studies based on this dataset. So it would be necessary to summarize the findings based on this dataset. This would help the readers to understand the importance of this dataset and leverage its key scientific implications.

Answer: The journal's requirements stated that: *Articles in the data section may pertain to the planning, instrumentation, and execution of experiments or collection of data.* ***Any interpretation of data is outside the scope of regular articles****. Articles on methods describe nontrivial statistical and other methods employed (e.g. to filter, normalize, or convert raw data to primary published data) as well as nontrivial instrumentation or operational methods. Any comparison to other methods is beyond the scope of regular articles.* Therefore, we focused on the data report, not focusing on the results. Also, as suggested, we have added the brief discussion on the data and their scientific aspects in each section (at the end of each sub-section).

Specific comments:

Line 390: strike through text

Answer: Revised.

Line 518: It's not necessary to mention 'which is of great importance to scientific communities worldwide' as it has been addressed in the introduction.

Answer: Agree and deleted.

---

## Author Response (AR2)

December 27, 2021

Dear Editor and the reviewer,

Thank all of you very much for taking the time and efforts to review our manuscript, originally titled with "APCC Data Report I: Black carbon and organic carbon dataset from atmosphere, glaciers, snow cover, precipitation, and lake sediment cores over the Tibetan Plateau and its surroundings", that we have submitted to "Earth System Science Data" (MS No.: essd-2021-187R1). We have considered all comments and suggestions carefully, and the title has been revised as "Black carbon and organic carbon dataset over the Third Pole". We also tried our best to address the related comments and revised the manuscript accordingly, and we hope the revised manuscript is suitable for publication in the journal.

Revisions are made to address the following specific comments.

Our point-by-point responses to the comments are listed below in blue color.

Yours sincerely,

Shichang Kang and Yulan Zhang on behalf of all co-authors

**Response to comments**

The referees agreed with the value of the dataset and were satisfied with the revision. After reading the manuscript, I believe that the previous comments and suggestions have been addressed. However, I still believe that minor updates can be made to further improve the manuscript:

1) I would suggest the authors give another thought on the title, which seems long, complicated, and possibly confusing, especially the expression "APCC Data Report". The ESSD manuscript focuses on data; hence, the submissions were expected to highlight datasets. I understand that the authors were presenting/reporting the first collection of the APCC data. However, there is a risk to be misunderstood as a report instead of peer-reviewed paper, particularly a data paper. Also, I do not think that APCC is a widely used abbreviation. Use of the term, especially in the title, could be confusing and misleading readers to think it as a report from IPCC or something.

Answer: Thank you very much for the suggestion. We considered the title again base on

your comment, and we agree that the title is too long. In the original title, use the abbreviation of "APCC" may be not clear to the article. Therefore, we revised the title as: Black carbon and organic carbon dataset over the Third Pole.

2) Line 123, change "10 years" to "decade" unless it referred to exact 10 years.

Answer: Sure, we have revised accordingly.

3) Line 138, I am not sure "domain" would be a proper term for the sub regions. I would suggest reconsidering it.

Answer: The term "domain" was borrowed from the previous study by Yao et al. (2013). In their study, they indicated that "The spatial and temporal patterns of precipitation $\delta^{18}O$ and their relationships with temperature and precipitation reveal three distinct domains, respectively associated with the influence of the westerlies (northern TP), Indian monsoon (southern TP), and transition in between." Considering it's rarely used, we use the term as you suggested as "sub regions".

References:

Yao T., Masson-Delmotte V., Gao J., et al., 2013. A review of climatic controls on $\delta^{18}O$ in precipitation over the Tibetan Plateau: observations and simulations. Rev. Geophys. Doi: 8755-1209/13/10.1002/rog.20023

4) The explanation in section 2.1 needs to be checked. It mentioned 29 stations (and sites) and then mentioned 138 sites were surveyed.

Answer: Stations in this study represent the continuous observation at this site. Usually, we setup the related instruments (e.g., TSP sampler, or Aethalometer AE33). However, "site" here is only referred to the locations we collected snow sample during our field work. There is no continuous observation performed. In order to make it clearer, we have tried to revise the explanation in the main text.

5) Section 2.2. I found that the explanation is hard to follow. I would suggest mention the

total site number at the beginning of the section, and then introduce the sites in sub regions.

Answer: Agree, we have added total site number as suggested.

6) Line 186, Chinese TP and its surroundings. The expression could be ambiguity. Did the authors mean the surrounding regions only in China?

Answer: Yes, in this sentence, we mean the station located in China. We have revised as "… continuously observed over the TP and its surroundings within China."

7) Conclusion section, I would suggest rewrite the first sentence of the first paragraph to focus on the presented dataset. It is a data description paper instead of publishing research.

Answer: The first sentence has been revised as:

The dataset of black carbon and organic carbon concentrations and their related MAC values and carbon isotope signatures from the atmosphere, glaciers, snow cover, precipitation, and lake sediments over the Third Pole region are presented.

8) The gray boundary line in figure 8 seem to be the boundary of China. I am not sure it was relevant to the figure. A similar suggestion to Figure 5 and 6, please only include country boundaries when it is relevant.

Answer: Agree, and we have revised the Figure 5, 6 and 8 as suggested.

9) Figure 11, please make sure the WSOC flux did not go off the chart. I would also suggest zoom further to the sites in the distribution map in Figure 11 to make it easier to recognize the site locations.

Answer: Agree, and we have revised.

Finally, grammar errors and ill expressions can still be found in the manuscript. For example:

10) Line 65, change "the earth's climate" to "Earth's climate" unless the authors referred to soil instead of the planet.

Answer: Revised.

11) Line 349, change "one thousand" to "1,000" or maybe "~1,000".

Answer: Revised.

12) Line 290, change "records data" to "records".

Answer: Revised.

13) Line 139, add "a" in front of "exact boundary".

Answer: Revised. Thank you very much again for all the comments and suggestion to improve our manuscript. We have tried to check the whole manuscript.